# CLUSTERED TASK-AWARE META-LEARNING BY LEARNING FROM LEARNING PATHS

## ABSTRACT

To enable effective learning of new tasks with only few samples, meta-learning acquires common knowledge from the existing tasks with a globally shared meta-learner. To further address the problem of task heterogeneity, recent developments balance between customization and generalization by incorporating task clustering to generate the task-aware modulation to be applied on the global meta-learner. However, these methods learn task representation mostly from the features of input data, while the task-specific optimization process with respect to the base-learner model is often neglected. In this work, we propose a **C**lustered **T**ask-**A**ware **M**eta-**L**earning (CTML) framework with task representation learned from both features and learning path. We first conduct rehearsed task learning from the common initialization, and collect a set of geometric quantities that adequately describes this learning path. By inputting this set of values into a meta path learner, we automatically abstract path representation optimized for the downstream clustering and modulation. To further save the computational cost incurred by the additional rehearsed learning, we devise a shortcut tunnel to directly map between the path and feature cluster assignments. Extensive experiments on two real-world application domains: few-shot image classification and cold-start recommendation demonstrate the superiority of CTML compared to state-of-the-art baselines.

## 1 INTRODUCTION

The astonishing performance of deep learning relies on large amounts of data, which are not always available. Humans, on the other hand, are able to learn new tasks much more quickly, leveraging prior experience to relate knowledge among tasks. Inspired by this property of human intelligence, meta-learning (also known as learning to learn) (Thrun & Pratt, 1998) acquires transferable commonalities from existing tasks for more efficient learning of new tasks. Although recent advances in meta-learning have demonstrated success in fields like few-shot image classification and cold-start recommendation (Vinyals et al., 2016; Snell et al., 2017; Ravi & Larochelle, 2016; Munkhdalai & Yu, 2017), most of them typically assume that all the tasks are drawn from a single distribution and face the challenge of handling tasks that come from different underlying distributions, a problem known as *task heterogeneity* (Devos & Dandi, 2020; Yao et al., 2020; Suo et al., 2020).

To overcome this challenge, many recently developed works leverage task-specific information to customize the global meta-learner (Vuorio et al., 2019; Oreshkin et al., 2018; Requeima et al., 2019; Finn et al., 2018; Yoon et al., 2018; Li et al., 2019; Rusu et al., 2018). To further consider generalization among related tasks, methods that perform various types of task clustering are proposed (Yao et al., 2019; 2020; Suo et al., 2020; Lu et al., 2020; Dong et al., 2020; Lin et al., 2021). Despite their effectiveness in improving over the globally shared meta-learning algorithms, they learn task representations only based on the input data distribution in (original or projected) feature space, while the interaction between data and the base-learner is often neglected.

The amount of information contained in the task-specific data about a network that is responsible for performing the task can be seen as a good representation of the task itself. This can be manifested as the gradients of the network parameters with respect to the task-specific loss, or Fisher Information Matrix (FIM) which indicates the importance of different network parameters in solving the task. Achille et al. (2019) introduce a task embedding method based on FIM to assist the meta task of selecting a pre-trained feature extractor. To alleviate the conflict issue of the global initialization methods, Baik et al. (2020) utilize task gradients at the initialization as task representation to produce

attenuation. However, these methods represent task based on gradients at only a single point in parameter space (e.g., at the initialization), while the potential of exploiting a wider range on the task manifold remains unexplored .

For the parameter initialization approach, the key of success lies in that the task adaptation process is accounted for when training the meta-learner. This task-specific learning may involve multiple gradient descent steps, and thus constitute a learning trajectory on the loss surface (Flennerhag et al., 2018; Zenke et al., 2017). To better characterize the task optimization behaviors, it is more beneficial to look at the complete learning trajectory as opposed to only the first gradient step at the initialization, as it is likely that tasks with similar gradients at first will have their learning paths diverged as the update proceeds. With that in mind, we are motivated to leverage the entire learning path for better task representation, which will then be used to condition the global initialization.

In particular, we propose a **C**lustered **T**ask-aware **M**eta-**L**earning (CTML) framework, building upon the well-known global initialization method Model-Agnostic Meta-Learning (MAML) (Finn et al., 2017). To address the problem of task heterogeneity with a good balance between customization and generalization, we modulate the common initialization based on task representation learned from both its local task-specific information and global clustering results. In addition to using the input features to represent task, we further leverage the task learning path with respect to the base-learner to characterize task from the perspective of optimization. To facilitate clustering, we devise a GRU-based meta path learner which automatically abstracts path representations from the step-wise geometric quantities. We realize that it may be too costly to rehearse the entire learning process for task representation. Hence, we further propose a shortcut tunnel to bypass the rehearsed task learning during meta-testing and predict path cluster assignment directly from the feature cluster assignment. We carefully study the effectiveness of CTML in two real-world application domains: few-shot image classification and cold-start recommendation, and show that our method is able to outperform the baselines with comparable inference time.

## 2 RELATED WORK

To address the problem of task heterogeneity, recent developments often tailor the shared meta-learner to different tasks using task-specific information. Finn et al. (2018) and Yoon et al. (2018) model the uncertainty exists in task distribution with probabilistic framework. Oreshkin et al. (2018), Li et al. (2019), Requeima et al. (2019) and Bateni et al. (2020) condition the base network on task-specific data by designing a meta adaptation network. To enable more robust training, Rusu et al. (2018) propose to learn a lower-dimensional latent space specific to each task to generate the base network parameters. Relevant to our work, Vuorio et al. (2019) build upon MAML and modulate the global initialization based on the task representation learned from an encoder network.

However, customizing the common knowledge to individual task without considering the relations among similar tasks may lead to poor generalization. In regard to this, Dong et al. (2020) and Lin et al. (2021) apply K-means clustering on users (treated as tasks) based on their profile information to address the cold-start problem in recommender systems. Yao et al. (2019) employ a hierarchical structure to model the nested relationships inherent in domains with clear taxonomy, such as image classification. Yao et al. (2020) further develop an automatic relational graph method by constructing a meta-knowledge graph to preserve and propagate the global structural information. Suo et al. (2020) take advantages of an external knowledge base to facilitate task clustering. Despite their effectiveness in generalizing across tasks, they rely solely on input features for task representation, while the interaction between task data and the base-learner (e.g., gradients) is often neglected.

The idea of using gradients for task representation is commonly adopted in continual learning (Kirkpatrick et al., 2017; Zenke et al., 2017; Aljundi et al., 2018). In the context of task-aware meta-learning, Achille et al. (2019) propose to use the FIM of a "probe" network to represent task, and Baik et al. (2020) leverage gradients of the globally initialized parameters to generate task-specific attenuation. However, these methods utilize gradients only at a single point in parameter space, whereas meta-learning often incorporates the entire task learning trajectory while learning the meta-learner. Garcia et al. (2021) perform task clustering at each adaptation step and aggregate the gradients for more stable task adaptation. Flennerhag et al. (2018) highlight the benefits of leveraging the learning paths for deriving the common knowledge, but the proposed algorithm only relies on one geometric quantity – the length of the path, which can be limited in characterizing the learning paths, and the initialization is not tailored to specific task. Our work serves to address the above-mentioned

limitations, whereby higher-order behaviours of the learning path are also taken into account to learn a better task representation for modulating the global initialization.

## 3 PRELIMINARIES AND PROBLEM STATEMENT

In task-heterogeneous setting, tasks $\{\mathcal{T}_1, \mathcal{T}_2, ..., \mathcal{T}_N\}$ are sampled from a mixture of task distributions $\{p_1(\mathcal{T}), p_2(\mathcal{T}), ...\}$, where the number of underlying distributions is usually unknown. The goal of meta-learning is to learn the sharable knowledge by training the meta-learner on a set of meta-training tasks $\mathbb{T}^{tr} = \{\mathcal{T}_i\}_{i=1}^{N^{tr}}$, and test it on a set of meta-testing tasks $\mathbb{T}^{te} = \{\mathcal{T}_i\}_{i=N^{tr}+1}^{N}$. For each task $\mathcal{T}_i \in \mathbb{T}^{tr} \cup \mathbb{T}^{te}$, its samples are further divided into a training set (also termed support set) $\mathcal{D}^{tr}_{\mathcal{T}_i} = \{(\mathbf{x}_{i,j}, y_{i,j})\}_{j=1}^{n^{tr}_{\mathcal{T}_i}}$ and a test set (also termed query set) $\mathcal{D}^{te}_{\mathcal{T}_i} = \{(\mathbf{x}_{i,j}, y_{i,j})\}_{j=n^{tr}_{\mathcal{T}_i}+1}^{n_{\mathcal{T}_i}}$, which together form an "episode" (Vinyals et al., 2016). This episodic scheme allows us to train tasks to learn fast during meta-training, and test the learning performance of new tasks in the same way during meta-testing. For few-shot learning, the size of the training set $n^{tr}_{\mathcal{T}_i}$ is usually small.

Our work builds upon Model-Agnostic Meta-Learning (MAML) (Finn et al., 2017). It implements the meta-learner as an initialization of parameters $\theta_0 \in \mathbb{R}^D$ of the base-learner $f_\theta$ responsible for the prediction task. During meta-training, the global initialization $\theta_0$ is first adapted to each meta-training task $\mathcal{T}_i \in \mathbb{T}^{tr}$ by learning on the respective training set $\mathcal{D}^{tr}_{\mathcal{T}_i}$, which yields the task-specific parameters $\theta_{\mathcal{T}_i}$. After that, loss is computed on the test set $\mathcal{D}^{te}_{\mathcal{T}_i}$ based on $\theta_{\mathcal{T}_i}$ and backward propagated to update $\theta_0$. Taking one step adaptation as an example, the meta-optimization is as follows:

$$\theta_0^* = \arg\min_{\theta_0} \sum_{\mathcal{T}_i \in \mathbb{T}^{tr}} \mathcal{L}(f_{\theta_{\mathcal{T}_i}}, \mathcal{D}^{te}_{\mathcal{T}_i}) = \arg\min_{\theta_0} \sum_{\mathcal{T}_i \in \mathbb{T}^{tr}} \mathcal{L}(f_{\theta_0 - \alpha\nabla_\theta \mathcal{L}(f_{\theta_0}, \mathcal{D}^{tr}_{\mathcal{T}_i})}, \mathcal{D}^{te}_{\mathcal{T}_i}), \quad (1)$$

where $\alpha$ is the adaptation rate, $\mathcal{L}(f_\theta, \mathcal{D})$ can be mean square error loss for regression problem (i.e., $\frac{1}{|\mathcal{D}|}\sum_{(\mathbf{x},y)\in\mathcal{D}}(y - f_\theta(\mathbf{x}))^2$), or cross-entropy loss for classification problem (i.e., $-\frac{1}{|\mathcal{D}|}\sum_{(\mathbf{x},y)\in\mathcal{D}} y \log f_\theta(\mathbf{x})$). During meta-testing, the learned initialization $\theta_0^*$ is adapted to each meta-testing task $\mathcal{T}_i \in \mathbb{T}^{te}$ using $\mathcal{D}^{tr}_{\mathcal{T}_i}$, and the learning performance is evaluated on $\mathcal{D}^{te}_{\mathcal{T}_i}$.

However, with a globally shared initialization, MAML is not capable of handling task heterogeneity. Though task-specic methods have been developed to tailor the global initialization, they lack explicit modeling of the global clustering structure. Hence, a framework considering both task-specific information and global structure is desired.

## 4 METHODOLOGY

Grounded on MAML, our CTML framework modulates the common initialization based on task representation learned from two different perspectives enhanced with the clustering information. Specifically, given the training set of a specific task, we first conduct rehearsed learning of the task from the common initialization and compute a set of step-wise quantities along the learning path. This set of values will be inputted into the meta path learner to generate a task-specific path embedding. On the other hand, input features of the given task will also be extracted to form the feature embedding. Both feature and path embeddings will then undergo soft K-means clustering in their respective latent space. Finally, the two embeddings enriched with cluster information will be aggregated to produce the task-aware modulation to be applied on the global initialization. From this modulated intialization, standard MAML follows, which conducts the actual task learning on the training set and perform meta-update across tasks using the test sets. To further improve the inference efficiency during meta-testing, a shortcut tunnel is used to bypass the rehearsed learning process and generate the path cluster assignment directly from the feature cluster assignment. An overview of CTML framework is shown in Figure 1.

In the following sections, we first elaborate on task representation learning based on learning path and features respectively, and then introduce the task-aware modulation and the shortcut tunnel.

### 4.1 TASK REPRESENTATION BASED ON LEARNING PATH

#### 4.1.1 PATH CONSTRUCTION

To obtain representation of task $\mathcal{T}_i$ based on learning path, we first conduct a $\tau$-step rehearsed learning from the global initialization $\theta_0$ on training set $\mathcal{D}^{tr}_{\mathcal{T}_i}$. Applying the same gradient descent update as in MAML, we obtain the updated parameters at each step $t \in \{1, 2, ..., \tau\}$ by:

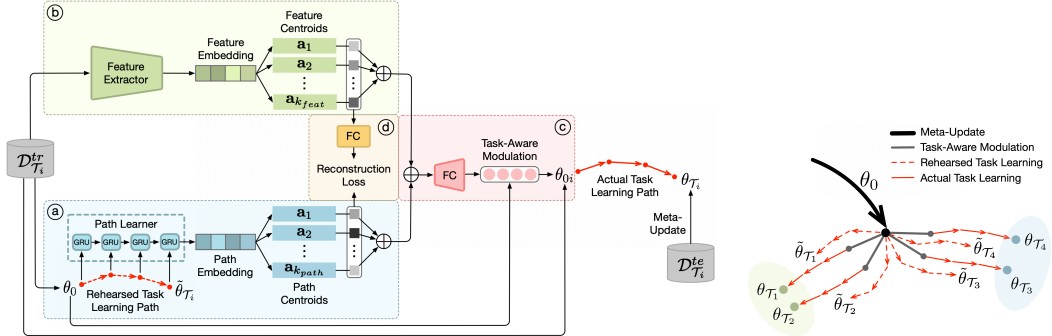

Figure 1: Overview of CTML framework. For each incoming task $\mathcal{T}_i$, part (a) extracts path representation and performs clustering; part (b) extracts feature representation and performs clustering; part (c) generates task-aware modulation for global initialization; part (d) reconstructs path cluster assignment from feature cluster assignment.

Figure 2: Illustration of CTML task learning process. Tasks with similar rehearsed learning paths will produce similar modulations.

$\tilde{\theta}^t_{\mathcal{T}_i} = \tilde{\theta}^{t-1}_{\mathcal{T}_i} - \alpha \nabla_\theta \mathcal{L}(f_{\tilde{\theta}^{t-1}_{\mathcal{T}_i}}, \mathcal{D}^{tr}_{\mathcal{T}_i})$, where $\tilde{\theta}^0_{\mathcal{T}_i} = \theta_0$. The overhead $\tilde{\cdot}$ is used to differentiate between the parameters updated from the rehearsed learning and from the actual learning.

Joining the discrete points $\{(\tilde{\theta}^t_{\mathcal{T}_i}, \tilde{\mathcal{L}}^t_{\mathcal{T}_i})\}^\tau_{t=0}$ constitutes the rehearsed learning path of $\mathcal{T}_i$ (we write $\mathcal{L}(f_{\tilde{\theta}^t_{\mathcal{T}_i}}, \mathcal{D}^{tr}_{\mathcal{T}_i})$ as $\tilde{\mathcal{L}}^t_{\mathcal{T}_i}$ for brevity) on the unique task manifold $M_{\mathcal{T}_i} \in \mathbb{R}^{D+1}$, characterized by the base-learner function and the task-specific data distribution. We collect the coordinates of point $(\tilde{\theta}^t_{\mathcal{T}_i}, \tilde{\mathcal{L}}^t_{\mathcal{T}_i})$ at each update step $t \in \{0, 1, 2, ..., \tau\}$ to indicate the exact learning trajectory. To account for higher-order behaviours of the learning path, we further incorporate the gradients $\nabla_\theta \tilde{\mathcal{L}}^t_{\mathcal{T}_i}$ and the Fisher Information Matrix (FIM) $\tilde{F}^t_{\mathcal{T}_i}$ at each step, which specify the direction and curvature respectively. Note that by definition, the FIM is equivalent to the Hessian matrix in expectation for cross-entropy loss: $\tilde{F}^t_{\mathcal{T}_i} = \mathbb{E}_{\mathbf{x},y \sim \mathcal{T}_i}[\nabla_\theta \mathcal{L}(f_{\tilde{\theta}^t_{\mathcal{T}_i}}, \mathbf{x}, y) \nabla_\theta \mathcal{L}(f_{\tilde{\theta}^t_{\mathcal{T}_i}}, \mathbf{x}, y)^\top] = \mathbb{E}_{\mathbf{x},y \sim \mathcal{T}_i}[\nabla^2_\theta \mathcal{L}(f_{\tilde{\theta}^t_{\mathcal{T}_i}}, \mathbf{x}, y)]$. It provides a measure of the amount of information that task $\mathcal{T}_i$ contains about the parameters $\theta$ (Gianfelici & Battistelli, 2009). Following Achille et al. (2019), we assume negligible correlation among different parameters and consider only the diagonal entries of $\tilde{F}^t_{\mathcal{T}_i}$, denoted by $\tilde{\mathcal{F}}^t_{\mathcal{T}_i}$. As a result, we obtain a set of step-wise quantities $\{(\tilde{\theta}^t_{\mathcal{T}_i}, \tilde{\mathcal{L}}^t_{\mathcal{T}_i}, \nabla_\theta \tilde{\mathcal{L}}^t_{\mathcal{T}_i}, \tilde{\mathcal{F}}^t_{\mathcal{T}_i})\}^\tau_{t=0}$ that adequately describes the rehearsed learning path.

As opposed to looking at only a single point in the parameter space (e.g., use gradients at the initialization to represent task (Baik et al., 2020)), it is benificial to take into account the entire learning path traversed, which gives a more complete picture of the optimization process. Take for instance, two tasks $\mathcal{T}_a$ and $\mathcal{T}_b$ may have gradients forming a small angle $\psi^0_{a,b}$ at the initialization, i.e., $\cos(\psi^0_{a,b}) > 0$. However, it is likely that their learning paths will diverge as the gradient update proceeds, i.e., the accumulated angle $\sum^\tau_{t=0} \cos(\psi^t_{a,b}) < 0$ (see dotted paths of $\mathcal{T}_2$ and $\mathcal{T}_3$ in Figure 2). Conversely, the tasks may have gradients pointing at different directions at first, but eventually converge towards the same direction as the learning proceeds (see dotted paths of $\mathcal{T}_1$ and $\mathcal{T}_2$). Considering a single step only can be restrictive for task representation from the optimization perspective. Looking further down the paths allows better characterization of the learning behaviors and even the flexibility to determine the "important" steps (as shown in experiments later), producing more informative task representations. Furthermore, leveraging path representations serves to provide a stronger inductive bias to condition the initialization in favorable direction, as the rehearsed learning path can be considered as encoding the quality of the initialization with respect to the task (i.e., tasks with longer and more zigzag paths imply that the initialization is not a good one for them).

### 4.1.2 PATH REPRESENTATION

Having collected the step-wise quantities $\{(\tilde{\theta}^t_{\mathcal{T}_i}, \tilde{\mathcal{L}}^t_{\mathcal{T}_i}, \nabla_\theta \tilde{\mathcal{L}}^t_{\mathcal{T}_i}, \tilde{\mathcal{F}}^t_{\mathcal{T}_i})\}^\tau_{t=0}$ along the rehearsed learning path, the problem now is how to evaluate "similarity" among different task learning paths for clustering. Previous works with single-step gradients usually compute dot product or cosine similarity between gradients of two tasks (Riemer et al., 2018; Katoch et al., 2019; Yu et al., 2020; Baik et al., 2020). With multiple steps, a straightforward adaptation will be to simply sum up the similarity scores at all steps. However, this human-defined rule may not be the best way of measuring path similarity. Instead, we propose to employ a meta path learner to automatically learn a

path embedding from the step-wise quantities, and then measure similarity based on these vector representations. In other words, the meta path learner induces a latent space on which the distance metric best characterizes what is considered as "similar" (or "disimilar") among task learning paths.

Before we delve into the design of the path learner, we first elaborate how we construct the functional input from the step-wise quantities. Recall that at each step $t$, we have $\tilde{\theta}^t_{\mathcal{T}_i}, \nabla_\theta \tilde{\mathcal{L}}^t_{\mathcal{T}_i}, \tilde{\mathcal{F}}^t_{\mathcal{T}_i} \in \mathbb{R}^D$ and $\tilde{\mathcal{L}}^t_{\mathcal{T}_i} \in \mathbb{R}$. To obtain a regularly shaped input, we duplicate $\tilde{\mathcal{L}}^t_{\mathcal{T}_i}$ to form a $D$-dimensional vector and stack it together with the other 3 components to form a matrix $\mathbf{P}^t_{\mathcal{T}_i} \in \mathbb{R}^{4 \times D}$. Further stacking the $\tau + 1$ steps gives the overall 3-D matrix $\mathbf{P}_{\mathcal{T}_i} \in \mathbb{R}^{(\tau+1) \times 4 \times D}$. To avoid high model complexity, we apply the meta path learner coordinate-wise on the base-learner parameters. That is, the same path learner is used to process the matrix $\mathbf{P}^d_{\mathcal{T}_i} \in \mathbb{R}^{(\tau+1) \times 4}$ independently for all $d \in \{1, 2, ..., D\}$.

For the path learner design, we propose to leverage the Gated Recurrent Units (GRUs) to model the sequential dependencies among steps[1]. Specifically, the hidden state $\mathbf{h}^{d,t}_i$ at step $t$ is obtained by inputting the $t$-th sliced vector $\mathbf{P}^d_{t,:} \in \mathbb{R}^4$ of $\mathbf{P}^d_{\mathcal{T}_i}$ and the hidden state $\mathbf{h}^{d,t-1}_i$ at step $t-1$ into the GRU cell:

$$\mathbf{r}^{d,t}_i = \sigma(\mathbf{W}_r \cdot [\mathbf{h}^{d,t-1}_i, \mathbf{P}^d_{t,:}]), \quad \mathbf{z}^{d,t}_i = \sigma(\mathbf{W}_z \cdot [\mathbf{h}^{d,t-1}_i, \mathbf{P}^d_{t,:}]),$$
$$\tilde{\mathbf{h}}^{d,t}_i = \tanh(\mathbf{W}_{\tilde{h}} \cdot [\mathbf{r}^{d,t}_i \odot \mathbf{h}^{d,t-1}_i, \mathbf{P}^d_{t,:}]), \quad \mathbf{h}^{d,t}_i = (1 - \mathbf{z}^{d,t}_i) \odot \mathbf{h}^{d,t-1}_i + \mathbf{z}^{d,t}_i \odot \tilde{\mathbf{h}}^{d,t}_i, \quad (2)$$

where $[\cdot, \cdot]$ denotes concatenation, $\sigma(\cdot)$ is the sigmoid activation, $\mathbf{r}^{d,t}_i, \mathbf{z}^{d,t}_i \in (0, 1)$ are gates that control how much of past and present information to be retained, and $\mathbf{W}_r, \mathbf{W}_z, \mathbf{W}_{\tilde{h}}$ are learnable weights shared across all steps. For step $t = 0$, we use a zero-initialized input hidden state.

The path representation $\mathbf{h}^d_i$ at the $d$-th dimension is obtained from the final step hidden state $\mathbf{h}^{d,\tau}_i$ via a linear transformation: $\mathbf{h}^d_i = \mathbf{W}_o \cdot \mathbf{h}^{d,\tau}_i + \mathbf{b}_o$. Concatenating the path representations at all $D$ dimensions and passing it through a fully-connected layer gives the final path embedding $\mathbf{e}^{path}_{\mathcal{T}_i} \in \mathbb{R}^{d_e}$ for task $\mathcal{T}_i$: $\mathbf{e}^{path}_{\mathcal{T}_i} = \text{FC}([\mathbf{h}^1_i, ..., \mathbf{h}^D_i]^\top)$.

Inspired by (Suo et al., 2020; Dong et al., 2020; Lin et al., 2021), we handle task heterogeneity without jeopardizing generalization among similar tasks by employing a simple yet effective soft K-means clustering on the path embeddings[2]. Specifically, we maintain $k_{path}$ learnable cluster centroids $\{\mathbf{a}^{path}_j | \forall j \in [1, k_{path}]\} \in \mathbb{R}^{k_{path} \times d_e}$ for path embeddings. The cluster-enhanced path embedding $\tilde{\mathbf{e}}^{path}_{\mathcal{T}_i} \in \mathbb{R}^{d_e}$ is simply the weighted sum of the cluster centroids:

$$\tilde{\mathbf{e}}^{path}_{\mathcal{T}_i} = \sum_{j=1}^{k_{path}} q^{path}_{ij} \mathbf{a}^{path}_j, \quad (3)$$

where $q^{path}_{ij} = \frac{(1 + ||\mathbf{e}^{path}_{\mathcal{T}_i} - \mathbf{a}^{path}_j||^2)^{-1}}{\sum_{j'} (1 + ||\mathbf{e}^{path}_{\mathcal{T}_i} - \mathbf{a}^{path}_{j'}||^2)^{-1}}$ is the probability of assigning $\mathbf{e}^{path}_{\mathcal{T}_i}$ to cluster $j$, computed using the Student's $t$-distribution as a kernel, following Xie et al. (2016).

## 4.2 TASK REPRESENTATION BASED ON FEATURES

Task representation based on learning path can be interpreted as encoding the conditional distribution $p(y|\mathbf{x})$ (as the learning path is informed by the labels), whereas task is best described by the joint distribution $p(\mathbf{x}, y) = p(y|\mathbf{x})p(\mathbf{x})$. Therefore, we further incorporate the marginal distribution of input features by learning another representation solely based on features, as what have been done in most of the existing task-aware meta-learning methods (Vuorio et al., 2019; Yao et al., 2019; 2020; Lin et al., 2021). Another significance of including feature-based representation is that, we can create a mapping between the path and feature representations to bypass the rehearsed learning during the meta-testing phase (details will be elaborated in Section 4.4).

Generally, the design of the feature extractor may vary for different application domains. Let $\mathcal{E}(\cdot)$ denote an arbitrary feature extractor, the feature embedding $\mathbf{e}^{feat}_{\mathcal{T}_i} \in \mathbb{R}^{d_e}$ for task $\mathcal{T}_i$ is obtained by aggregating the extracted features of all samples in the training set: $\mathbf{e}^{feat}_{\mathcal{T}_i} = \frac{1}{n^{tr}_{\mathcal{T}_i}} \sum_{j=1}^{n^{tr}_{\mathcal{T}_i}} (\mathcal{E}(\mathbf{x}_{i,j}))$.

Similar to the path embeddings, we also employ a soft K-means clustering to promote generalization among related feature embeddings. Specifically, given $k_{feat}$ cluster centroids $\{\mathbf{a}^{feat}_j | \forall j \in$

---

[1]Other network designs are possible for the meta path learner. We compare their efficacy in ablation study.
[2]Other clustering algorithms are possible. We leave it to future work.

$[1, k_{feat}]\} \in \mathbb{R}^{k_{feat} \times d_e}$, the cluster-enhanced feature embedding $\tilde{\mathbf{e}}_{\mathcal{T}_i}^{feat} \in \mathbb{R}^{d_e}$ is obtained by:

$$\tilde{\mathbf{e}}_{\mathcal{T}_i}^{feat} = \sum_{j=1}^{k_{feat}} q_{ij}^{feat} \mathbf{a}_j^{feat}, \qquad (4)$$

where $q_{ij}^{feat} = \frac{(1+||\mathbf{e}_{\mathcal{T}_i}^{feat} - \mathbf{a}_j^{feat}||^2)^{-1}}{\sum_{j'}(1+||\mathbf{e}_{\mathcal{T}_i}^{feat} - \mathbf{a}_{j'}^{feat}||^2)^{-1}}$ is the probability of assigning $\mathbf{e}_{\mathcal{T}_i}^{feat}$ to cluster $j$.

### 4.3 TASK-AWARE MODULATION

We aggregate the path and feature embeddings via a learnable weight vector $\lambda$ to generate the final task representation. This task representation will then be used to produce a modulation to tailor the global initialization $\theta_0$ to specific task. The modulated initialization $\theta_{0i}$ for task $\mathcal{T}_i$ is obtained by:

$$\theta_{0i} = \sigma(\mathbf{W} \cdot (\lambda \tilde{\mathbf{e}}_{\mathcal{T}_i}^{path} \oplus (1-\lambda)\tilde{\mathbf{e}}_{\mathcal{T}_i}^{feat}) + \mathbf{b}) \odot \theta_0, \qquad (5)$$

where $\mathbf{W}, \mathbf{b}, \lambda$ are learnable parameters, $\sigma(\cdot)$ is the sigmoid function, $\oplus$ and $\odot$ denote element-wise addition and element-wise multiplication respectively.

From this modulated initialization, task $\mathcal{T}_i$ will undergo regular task adaptation learning for $\tau$ steps. After that, the meta-learner $\phi$ will be updated by optimizing loss across all the test sets, the same procedure as in MAML. The meta-optimization objective is:

$$\min_\phi \sum_{\mathcal{T}_i \in \mathbb{T}^{tr}} \mathcal{L}(f_{\theta_{0i} - \alpha \sum_{t=0}^{\tau-1} \nabla_\theta \mathcal{L}(f_{\theta_{\mathcal{T}_i}^t}, \mathcal{D}_{\mathcal{T}_i}^{tr})}, \mathcal{D}_{\mathcal{T}_i}^{te}), \qquad (6)$$

where $\phi$ includes the global initialization $\theta_0$, the learnable parameters used to generate the cluster-enhanced path and feature embeddings, and also the final modulation.

### 4.4 IMPROVING META-TESTING EFFICIENCY VIA SHORTCUT TUNNEL

The need to conduct rehearsed learning before the actual learning of each task inevitably leads to twice the inference time compared to the original MAML. Though it is not possible to bypass the rehearsed learning during meta-training as we need to optimize the meta path learner and the downstream cluster centroids, it is possible to improve the inference efficiency during meta-testing with the well-trained cluster centroids. Note that the cluster-enhanced path embedding $\tilde{\mathbf{e}}_{\mathcal{T}_i}^{path}$ used to generate modulation is obtained based on the path cluster centroids and the soft assignment (see equation 3). If we can estimate the path cluster assignment without actually going through the rehearsed learning process to obtain $\mathbf{e}_{\mathcal{T}_i}^{path}$, we will be able to reduce the inference time by half.

Inspired by this, we devise a shortcut tunnel to predict the path cluster assignment directly from the feature cluster assignment of the same task. The assumption behind is that there exists a one-to-one mapping between the feature cluster assignments and the path cluster assignments, which can be linear or non-linear. For better expressivity, we employ a two-layer fully connected network to approximate the mapping and use it to reconstruct the path cluster assignment of $\mathcal{T}_i$ from its feature cluster assignment:

$$\hat{\mathbf{q}}_i^{path} = \text{softmax}(\text{FCs}(\mathbf{q}_i^{feat})), \qquad (7)$$

where $\mathbf{q}_i^{feat} = [q_{i1}^{feat}, ..., q_{ik_{feat}}^{feat}]^\top \in \mathbb{R}^{k_{feat}}$ and $\hat{\mathbf{q}}_i^{path} = [\hat{q}_{i1}^{path}, ..., \hat{q}_{ik_{path}}^{path}]^\top \in \mathbb{R}^{k_{path}}$. The mapping handles cases where $k_{path} \neq k_{feat}$ and allows for different softness of cluster assignments. To align the reconstructed and the actual assignment distributions, we use the Jensen-Shannon (JS) divergence (a symmetrized version of Kullback-Leibler (KL) divergence) as the reconstruction loss:

$$\mathcal{L}_r(\mathcal{D}_{\mathcal{T}_i}^{tr}) = \text{JS}(\hat{\mathbf{q}}_i^{path}||\mathbf{q}_i^{path}) = \frac{1}{2}\text{KL}(\hat{\mathbf{q}}_i^{path}||\mathbf{p}_i) + \frac{1}{2}\text{KL}(\mathbf{q}_i^{path}||\mathbf{p}_i), \qquad (8)$$

where $\mathbf{p}_i = \frac{1}{2}(\hat{\mathbf{q}}_i^{path} + \mathbf{q}_i^{path})$, and $\text{KL}(\mathbf{q}||\mathbf{p}) = \sum_j q_j \log \frac{q_j}{p_j}$ is the KL divergence. During meta-training, the reconstruction loss is optimized together with the loss in equation 6, resulting in the following overall objective:

$$\min_\phi \sum_{\mathcal{T}_i \in \mathbb{T}^{tr}} \mathcal{L}(f_{\theta_{0i} - \alpha \sum_{t=0}^{\tau-1} \nabla_\theta \mathcal{L}(f_{\theta_{\mathcal{T}_i}^t}, \mathcal{D}_{\mathcal{T}_i}^{tr})}, \mathcal{D}_{\mathcal{T}_i}^{te}) + \zeta \mathcal{L}_r(\mathcal{D}_{\mathcal{T}_i}^{tr}), \qquad (9)$$

where $\zeta$ controls the weight of $\mathcal{L}_r(\mathcal{D}_{\mathcal{T}_i}^{tr})$, and $\phi$ now further includes the shortcut tunnel parameters.

To apply the shortcut tunnel during meta-testing, we simply replace $\mathbf{q}_i^{path}$ with $\hat{\mathbf{q}}_i^{path}$ in equation 3 to obtain the cluster-enhanced path embedding $\tilde{\mathbf{e}}^{path}$, i.e., $\tilde{\mathbf{e}}_{\mathcal{T}_i}^{path} = \sum_{j=1}^{k_{path}} \hat{q}_{ij}^{path} \mathbf{a}_j^{path}$. The overall meta-training and meta-testing procedures of CTML are summarized in Algorithm 1 and 2.

**Algorithm 1:** Meta-Training of CTML

**Required:** Meta-training tasks $\mathbb{T}^{tr} = \{\mathcal{T}_i\}_{i=1}^{N^{tr}}$;
  Number of steps $\tau$; Adaptation rate $\alpha$;
  Meta-update rate $\beta$; Number of clusters
  $k_{path}$ and $k_{feat}$

1 Randomly initialize all learnable parameters $\phi$
2 **while** *not converged* **do**
3     Sample a batch of tasks $\mathcal{B}$ from $\mathbb{T}^{tr}$
4     **for** $\mathcal{T}_i \in \mathcal{B}$ **do**
5         Extract $\mathbf{e}_{\mathcal{T}_i}^{feat}$ and cluster to obtain $\tilde{\mathbf{e}}_{\mathcal{T}_i}^{feat}$
6         Extract $\mathbf{e}_{\mathcal{T}_i}^{path}$ from the $\tau$-step rehearsed
          learning and cluster to obtain $\tilde{\mathbf{e}}_{\mathcal{T}_i}^{path}$
7         Aggregate $\tilde{\mathbf{e}}_{\mathcal{T}_i}^{feat}$ and $\tilde{\mathbf{e}}_{\mathcal{T}_i}^{path}$ to generate $\theta_{0i}$
8         Evaluate $\mathcal{L}(f_{\theta_{\mathcal{T}_i}}, \mathcal{D}_{\mathcal{T}_i}^{te})$ and $\mathcal{L}_r(\mathcal{D}_{\mathcal{T}_i}^{tr})$
9     Meta-update $\phi \leftarrow$
      $\phi - \beta \nabla_\phi \sum_{\mathcal{T}_i \in \mathcal{B}}[\mathcal{L}(f_{\theta_{\mathcal{T}_i}}, \mathcal{D}_{\mathcal{T}_i}^{te}) + \zeta \mathcal{L}_r(\mathcal{D}_{\mathcal{T}_i}^{tr})]$

**Algorithm 2:** Meta-Testing of CTML

**Required:** Meta-testing tasks $\mathbb{T}^{te} = \{\mathcal{T}_i\}_{i=N_{\mathcal{T}_i}^{tr}+1}^{N}$;
  Number of steps $\tau$; Adaptation rate $\alpha$;
  Number of clusters $k_{path}$ and $k_{feat}$;
  Meta-trained $\phi$

1 **for** $\mathcal{T}_i \in \mathbb{T}^{te}$ **do**
2     Extract $\mathbf{e}_{\mathcal{T}_i}^{feat}$ and cluster to obtain $\tilde{\mathbf{e}}_{\mathcal{T}_i}^{feat}$
3     **if** *use shortcut approximation* **then**
4         Generate path assignment via the shortcut
          tunnel to obtain $\tilde{\mathbf{e}}_{\mathcal{T}_i}^{path}$
5     **else**
6         Extract $\mathbf{e}_{\mathcal{T}_i}^{path}$ from the $\tau$-step rehearsed
          learning and cluster to obtain $\tilde{\mathbf{e}}_{\mathcal{T}_i}^{path}$
7     Aggregate $\tilde{\mathbf{e}}_{\mathcal{T}_i}^{feat}$ and $\tilde{\mathbf{e}}_{\mathcal{T}_i}^{path}$ to generate $\theta_{0i}$
8     Evaluate $\mathcal{L}(f_{\theta_{\mathcal{T}_i}}, \mathcal{D}_{\mathcal{T}_i}^{te})$

## 5 EXPERIMENTS

To test the effectiveness of the proposed CTML framework, we conduct experiments[3] on two real-world application domains: few-shot image classification and cold-start recommendation.

### 5.1 FEW-SHOT IMAGE CLASSIFICATION

**Datasets and Settings** In few-shot image classification, each task is defined as assigning images to $N$ classes after training with $K$ samples (i.e., $N$-way $K$-shot) (Vinyals et al., 2016). To simulate task heterogeneity, we construct the Mixture-Of-Datasets consisting of 6 widely used benchmark datasets: CUB-200-2011 (Bird) (Wah et al., 2011), FGVC-Aircraft (Aircraft) (Maji et al., 2013), FGVCx-Fungi (Fungi) (Schroeder & Cui, 2018), VGG Flower (Flower) (Nilsback & Zisserman, 2008), Describable Textures (Texture) (Cimpoi et al., 2014), and GTSRB Traffic Signs (Traffic Sign) (Stallkamp et al., 2012). Following Yao et al. (2019), we create each task by sampling $N$ classes from one of the 6 sub-datasets. This step ensures that tasks are drawn from different underlying distributions. Apart from the Mixture-Of-Datasets, we also conduct experiments on the MiniImagenet (Ravi & Larochelle, 2016), which can be considered as a task-homogeneous setting as tasks are sampled from a single dataset (results on MiniImagenet are discussed in Appendix D.2). More details on datasets and pre-processing can be found in Appendix A.1.

**Baselines and Our Method** We compare the performance of our method against 5 baselines: (1) global initialization method: MAML (Finn et al., 2017); (2) task-specific initialization methods: MMAML (feature-based customization) (Vuorio et al., 2019) and L2F (gradient-based customization) (Baik et al., 2020); (3) cluster-enhanced initialization methods: HSML (feature-based hierarchical clustering) (Yao et al., 2019) and ARML (feature-based relational graph) (Yao et al., 2020). For our proposed CTML, we report both the original meta-testing performance (CTML) and the one applying shortcut approximation (CTML(approx.)). We further implement 2 variants to compare with the baselines: CTML-feat which only depends on feature representation and CTML-path which only depends on path representation. Following Finn et al. (2017), we adopt both the feature extractor $\mathcal{E}(\cdot)$ and the base-learner $f_\theta$ as a four-layer $3 \times 3$ convolutions with $32$ filters for all the compared methods (experiments on deeper network for base-learner can be found in Appendix D.1). Further details on hyper-parameters settings can be found in Appendix B.1.

**Baseline Comparison** Table 1a presents the 5-way 1-shot & 5-shot classification performance on Mixture-Of-Datasets (disentangled performance on individual sub-datasets are included in Appendix D.3). First of all, we see that all the task-aware methods outperform the globally initialized MAML in task-heterogeneous setting. L2F with gradient-based conditioning on the initialization performs better than the feature-based MMAML. After considering generalization across similar tasks with some clustering techniques, HSML and ARML further improve over MMAML. Under our proposed CTML framework, CTML-feat with K-means clustering on features exhibits comparable performance as HSML and ARML. For CTML-path, the improvements over baselines are more

---

[3]All experiments are conducted using NVIDIA Tesla P100 GPU with 16GB memory.

Table 1: Few-shot classification performance on Mixture-Of-Datasets for (a) baseline comparison and (b) ablation study. We sampled 1000 tasks for meta-testing. The results are reported in the form of mean accuracy (%) $\pm$ std over 8 trials. We also report the inference time (in milliseconds) per task during meta-testing for the 5-way 1-shot scenario for baseline comparison.

(a) Baseline Comparison

| Methods | Mixture-Of-Datasets | | |
| | 5-way 1-shot | 5-way 5-shot | runtime |
| --- | --- | --- | --- |
| MAML | $52.83 \pm 1.54$ | $66.54 \pm 0.61$ | 82.9 |
| MMAML | $54.28 \pm 1.30$ | $67.82 \pm 0.59$ | 83.2 |
| L2F | $56.61 \pm 1.21$ | $68.45 \pm 0.54$ | 117.9 |
| HSML | $56.32 \pm 1.16$ | $68.73 \pm 0.41$ | 101.3 |
| ARML | $57.21 \pm 1.68$ | $69.29 \pm 0.42$ | 93.0 |
| CTML-feat | $57.14 \pm 1.38$ | $68.71 \pm 0.52$ | 84.3 |
| CTML-path | $57.74 \pm 0.67$ | $70.63 \pm 0.68$ | 171.2 |
| CTML | $\mathbf{59.03 \pm 0.84}$ | $\mathbf{72.18 \pm 0.42}$ | 181.0 |
| CTML(approx.) | $58.91 \pm 0.95$ | $71.69 \pm 0.61$ | 85.7 |

(b) Ablation Study

| Variants | Mixture-Of-Datasets | |
| | 5-way 1-shot | 5-way 5-shot |
| --- | --- | --- |
| Remove $\tilde{\theta}^t_{\mathcal{T}_i}$ | $56.74 \pm 1.03$ | $71.38 \pm 0.52$ |
| Remove $\tilde{\mathcal{L}}^t_{\mathcal{T}_i}$ | $57.24 \pm 1.20$ | $71.50 \pm 0.71$ |
| Remove $\nabla_\theta \tilde{\mathcal{L}}^t_{\mathcal{T}_i}$ | $57.89 \pm 1.33$ | $71.76 \pm 0.61$ |
| Remove $\tilde{\mathcal{F}}^t_{\mathcal{T}_i}$ | $58.13 \pm 1.12$ | $71.88 \pm 0.67$ |
| Linear Path Learner | $57.52 \pm 1.09$ | $71.44 \pm 0.47$ |
| FC Path Learner | $58.37 \pm 1.02$ | $71.62 \pm 0.39$ |
| Attention Path Learner | $58.67 \pm 1.40$ | $71.83 \pm 0.56$ |
| No Clustering | $57.29 \pm 1.24$ | $71.06 \pm 0.62$ |
| CTML | $\mathbf{59.03 \pm 0.84}$ | $\mathbf{72.18 \pm 0.42}$ |

Figure 3: Understanding the path learning mechanism. (a) 6 tasks (5-way 1-shot) are randomly sampled from Aircraft, Flower and Traffic Sign sub-datasets. (b) PCA visualization of the 5-step rehearsed learning paths (upper plot) and the corresponding path embeddings generated by the GRU (lower plot) for the 6 tasks. (c) Visualization of z gate (how much to retain for the current step input) and r gate (how much to retain for the previous step memory) of GRU at each step $t$ for the 6 tasks.

significant, especially for the 5-shot scenario where the learning paths are more reliable with more training data. The final CTML combining the advantages of the two representations achieves further improvements and outperforms the strongest baseline by a significant margin. Nevertheless, we can see that the meta-testing time of CTML is about twice that of the baselines due to the additional rehearsed learning process. After applying the shortcut approximation, we are able to cut down the inference time by half with just a small compromise on performance.

**Ablation Study** We further conduct ablation study to test the efficacy of various CTML designs, including the step-wise input components, the path learner design, and the effects of clustering. From the results in Table 1b, we see that the lower-order quantities seem more important. Regarding the path learner design, our proposed GRU-based model achieves the best result due to its ability to capture the sequential dependencies among the steps. Attention model is also a competitive alternative, as the positional encodings and the causal mask serve to constrain the sequential order (detailed descriptions of the different path learner designs can be found in Appendix C). As for clustering, we see that it indeed helps to generalize better as the global structure is explicitly considered.

**Understanding the Path Learning Mechanism** To understand what has actually been learned by the GRU path learner, we randomly sample 6 tasks (5-way 1-shot) from Aircraft, Flower and Traffic Sign sub-datasets (Figure 3a) and visualize their path learning results. In Figure 3b, we plot the 5-step rehearsed learning paths (upper) and the corresponding path embeddings generated by the GRU (lower) on the first two PCA components. We can see that the learning paths of tasks from the same sub-dataset generally move along the same direction, and the corresponding path embeddings are able to encode the path geometric information and positioned closely for similar tasks in the latent space. In Figure 3c, we further visulize the z gates and r gates of the GRU at each step $t$, which control how much to retain for the current step input and previous step memory respectively. We can see that the GRU path learner is able to identify the "important" steps (darker-blue blocks) and absorbs a larger portion of the inputs at these steps into the final path embedding. More examples on path visualization can be found in Appendix D.5.

Table 2: Prediction performance in MAE (lower the better) for 3 datasets. We report the mean MAE over 5 trials, where the standard deviations are all less than 0.001.

| Methods | MovieLens | Yelp | Amazon |
|---|---|---|---|
| MeLU | 0.8164 | 0.9464 | 0.7216 |
| PAML | 0.7725 | 0.9280 | 0.7150 |
| MAMO | 0.8084 | 0.8965 | 0.7207 |
| MetaHIN | 0.7727 | 0.8832 | 0.6743 |
| CTML-feat | 0.7560 | 0.9023 | 0.6804 |
| CTML-path | 0.7592 | 0.8714 | 0.6286 |
| CTML | **0.7543** | **0.8603** | **0.6048** |
| CTML(approx.) | 0.7555 | 0.8679 | 0.6087 |

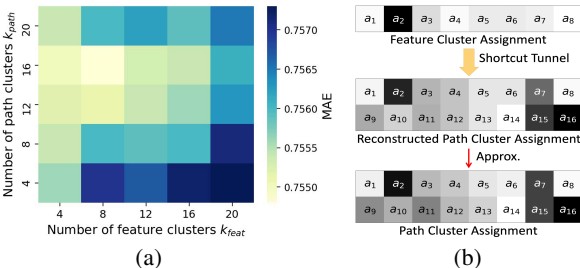

Figure 4: (a) Effect of varying $k_{feat}$ and $k_{path}$ for MovieLens dataset. (b) Shortcut approximation of path cluster assignment for a user from MovieLens dataset, where $k_{feat} = 8$ and $k_{path} = 16$.

## 5.2 COLD-START RECOMMENDATION

**Datasets and Settings** In the field of recommender systems, meta-learning has been applied to solve the cold-start problem, which refers to making recommendations related to new items or new users unseen before (Vartak et al., 2017; Volkovs et al., 2017; Lee et al., 2019). Here, we focus on the user cold-start problem, where recommending for each user is treated as a task. This setting is task-heterogeneous in nature as users may belong to different preference groups. We conduct experiments on 3 public datasets: MovieLens-1M (Harper & Konstan, 2015), Yelp (Dataset, 2019), and Amazon-CDs (McAuley et al., 2015), consisting of user's ratings on movies, business services and CD products respectively. We split each dataset into meta-train/val/test by ratio 7/1/2 according to the timestamps of ratings. For each user, we use the first 10 samples as training set, and the rest as test set. Further details about datasets and pre-processing can be found in Appendix A.2.

**Baselines and Our Method** We choose 4 baselines that specifically tackle the user cold-start problem in recommender systems developed based on MAML. MeLU (Lee et al., 2019) is the first to adopt MAML in recommender systems with locally adapted decision layers; MetaHIN (Lu et al., 2020) further incorporates Heterogeneous Information Networks (HIN) for data augmentation and multi-faceted adaptations; MAMO (Dong et al., 2020) involves a global memory module to group users based on profile information and personalizes the initial bias; PAML (Yu et al., 2021) personalizes the adaptation learning rate to allow for better fitting of the minor users. Following the baselines, we implement the feature extractor $\mathcal{E}(\cdot)$ as several embedding lookup matrices for different features, and the base-learner $f_\theta$ as a general recommender system model consists of embedding matrices followed by multiple fully-connected layers (Lee et al., 2019). Detailed hyper-parameters settings can be found in Appendix B.2.

**Baseline Comparison** We evaluate the rating prediction performance in terms of mean absolute error (MAE) on 3 datasets, reported in Table 2 (more detailed results can be found in Appendix D.4). From the results, we can see that PAML and MAMO with personalized adaptation rate and initialization achieve better performance than MELU. Our CTML surpasses all the baselines, including MetaHIN which leverages HIN for data augmentation. Furthermore, we notice that for MovieLens, CTML-feat is slightly better than CTML-path, while for the other two datasets, CTML-path significantly outperforms CTML-feat. Given that the MovieLens dataset contains more complete user profile, this result implies the potential of representing users based on learning paths when the side information is seriously lacking or inaccurate, which is often the case in real-world applications.

**Number of Clusters** For recommendation problem, there is no ground-truth grouping of users. In Figure 4a, we investigate the effect of varying the number of clusters $k_{path}$ and $k_{feat}$ for MovieLens dataset. We see that the best performance occurs at smaller $k_{feat}$ values and larger $k_{path}$ values, which implies greater complexity for clustering the learning paths. Our design of the shortcut tunnel allows flexibility to set different $k_{path}$ and $k_{feat}$. In Figure 4b, we visualize the shortcut approximation of path assignment from feature assignment for a random user of MovieLens. We can see that the shortcut tunnel is rather reliable as the reconstructed path assignment highly resembles the actual path assignment. More examples on shortcut approximation are provided in Appendix D.6.

## 6 CONCLUSION

In this work, we introduce a CTML framework which leverages both features and learning paths for task representation. We employ a GRU-based meta path learner to process the step-wise geometric quantities, and introduce a shortcut tunnel to bypass the rehearsed learning during meta-testing. Experiments on two real-world application domains demonstrate the effectiveness of our framework.

## 7 REPRODUCIBILITY STATEMENT

Our code can be accessed at:

https://github.com/didiya0825/ctml_code

For data pre-processing and hyper-parameters settings, please see Appendix A and B.

## 8 ETHICS STATEMENT

Since the proposed method is a general meta-learning framework, we do not foresee any potential ethic issue to the best our knowledge.

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

# A    DATASETS AND PRE-PROCESSING DETAILS

If not specially mentioned, datasets are downloaded from the respective official websites, where we have tried our best to seek permission for usage if applicable. The open-source datasets are licensed under CC0: Public Domain. For recommendation datasets, all records are anonymized and no personal identifiable information is involved.

## A.1    IMAGE DATASETS

In our experiments, Mixture-Of-Datasets consist of six widely used benchmark datasets. For the CUB-200-2011 (Bird), FGVC-Aircraft (Aircraft), FGVCx-Fungi (Fungi), and DTD (Texture) datasets, we directly acquire the pre-processed version released by the authors of HSML[4] (Yao et al., 2019). Pre-processing details of these four sub-datasets can be found in the original paper of HSML. We have obtained the authors' consent to use this asset for our research.

For the VGG Flower (Flower)[5] and GTSRB (Traffic Sign)[6] datasets, we download the raw datasets from the respective official websites. We follow the pre-processing protocols of Finn et al. (2017) to resize all the images to $84 \times 84 \times 3$. We also try to maintain the same meta-train/validation/test ratio as Yao et al. (2019) for class splits. For VGG Flower, there are in total 102 flower classes, and each class consists of 40 to 258 images. For GTSRB, we use the official training data which contains 43 traffic sign classes, and each class has 30 images.

For MiniImagenet, we acquire the dataset as well as the meta-train/validation/test splits from Ravi & Larochelle (2016). Table 3 summarizes the meta-train/validation/test splits for each of the sub-datasets as well as MiniImagenet.

Table 3: Meta-train/validation/test splits of sub-datasets and MiniImagenet.

| Datasets | #Meta-Train Classes | #Meta-Validation Classes | #Meta-Test Classes |
|---|---|---|---|
| CUB-200-2011 (Bird) (Wah et al., 2011) | 64 | 16 | 20 |
| FGVC-Aircraft (Aircraft) (Maji et al., 2013) | 64 | 16 | 20 |
| FGVCx-Fungi (Fungi) (Schroeder & Cui, 2018) | 64 | 16 | 24 |
| VGG Flower (Flower) (Nilsback & Zisserman, 2008) | 64 | 16 | 22 |
| DTD (Texture) (Cimpoi et al., 2014) | 30 | 7 | 10 |
| GTSRB (Traffic Sign) (Stallkamp et al., 2012) | 26 | 7 | 10 |
| MiniImagenet (Ravi & Larochelle, 2016) | 64 | 20 | 16 |

## A.2    RECOMMENDATION DATASETS

### A.2.1    MOVIELENS-1M

MovieLens-1M[7] dataset (Harper & Konstan, 2015) contains 1,000,209 anonymous ratings on around 3,900 movies made by 6,040 users. We acquire the additional side information released by the authors of MeLU[8] (Lee et al., 2019), which contains user profile information of gender, age group, occupation, zipcode, and movie profile information of rate, genre, director and actor.

The dataset is split into meta-train/validation/test set according to the timestamps of ratings by ratio 7/1/2. As a result, the start time is 2000-04-26, the meta-train cut-off time is 2000-11-22, the meta-validation cut-off time is 2000-12-02, and the end time is 2003-03-01. Users with less than 30 samples in each split are removed. In the end, the average sample size for each user is 163. The statistics of the pre-processed MovieLens-1M dataset and the number of meta-train/validation/test users are recorded in Table 4 and 5. Note that it is possible for meta-train users to appear in meta-validation and meta-test set, and these users are termed **warm users**. Conversely, meta-validation or meta-test users that have never appeared in the meta-train set are termed **cold users**.

---

[4]https://github.com/huaxiuyao/HSML
[5]https://www.robots.ox.ac.uk/~vgg/data/flowers/
[6]https://benchmark.ini.rub.de/
[7]https://grouplens.org/datasets/movielens/
[8]https://github.com/hoyeoplee/MeLU

### A.2.2  YELP

Yelp[9] dataset (Dataset, 2019) contains 8,635,403 reviews for 160,585 business services made by 2,189,457 users. We extract the star ratings from the reviews for our experiments. For each business service, we collect the city it resides and the categories it belongs to from the business metadata. However, for the user, the personal information is lacking, and we only have access to the user ID.

The dataset is split into meta-train/validation/test set according to the timestamps of ratings by ratio 7/1/2. As a result, the start time is 2004-10-12, the meta-train cut-off time is 2018-02-07, the meta-validation cut-off time is 2018-09-04, and the end time is 2019-12-13. Users with less than 30 samples in each split are removed. In the end, the average sample size for each user is 89. The statistics of the pre-processed Yelp dataset and the number of meta-train/validation/test users are recorded in Table 4 and 5.

### A.2.3  AMAZON-CDS

Amazon-CDs[10] dataset (McAuley et al., 2015) contains 3,749,004 ratings for 544,442 CD and Vinyl products made by 1,578,597 users. For each CD, we collect its brand and category, while for the user, the personal profile information is lacking and we only have access to the user ID.

The dataset is split into meta-train/validation/test set according to the timestamps of ratings by ratio 7/1/2. As a result, the start time is 1997-09-13, the meta-train cut-off time is 2012-01-13, the meta-validation cut-off time is 2013-03-05, and the end time is 2014-07-23. Users with less than 30 samples in each split are removed. In the end, the average sample size for each user is 82. The statistics of the pre-processed Amazon-CDs dataset and the number of meta-train/validation/test users are recorded in Table 4 and 5.

Table 4: Statistics of recommendation datasets.

| Datasets | Features | #IDs |
|---|---|---|
| MovieLens-1M (Harper & Konstan, 2015) | User | 5,255 |
| | Gender | 2 |
| | Age Group | 7 |
| | Occupation | 21 |
| | Zipcode | 3,402 |
| | Movie | 3,697 |
| | Rate | 6 |
| | Genre | 25 |
| | Director | 2,186 |
| | Actor | 8,030 |
| Yelp (Dataset, 2019) | User | 25,206 |
| | Business Service | 159,611 |
| | Category | 1,317 |
| | City | 1,023 |
| Amazon-CDs (McAuley et al., 2015) | User | 3,876 |
| | CD | 132,209 |
| | Category | 490 |
| | Brand | 49,283 |

Table 5: Number of meta-train/validation/test users of recommendation datasets.

| Datasets | #Meta-Train Users | #Meta-Validation Users | | | #Meta-Test Users | | |
|---|---|---|---|---|---|---|---|
| | | All | Warm | Cold | All | Warm | Cold |
| MovieLens-1M (Harper & Konstan, 2015) | 4,119 | 621 | 122 | 499 | 1,229 | 485 | 744 |
| Yelp (Dataset, 2019) | 22,822 | 1,281 | 681 | 600 | 3,442 | 1,441 | 2,001 |
| Amazon-CDs (McAuley et al., 2015) | 3,584 | 202 | 110 | 92 | 355 | 119 | 236 |

---

[9]https://www.yelp.com/dataset
[10]https://jmcauley.ucsd.edu/data/amazon/

# B  HYPER-PARAMETERS SETTINGS

## B.1  FEW-SHOT IMAGE CLASSIFICATION

For fair comparison, we adopt the same feature extractor and base-learner for all the compared methods. Following Finn et al. (2017), the feature extractor is a four-layer $3 \times 3$ convolutions with 32 filters, followed by a one-layer fully-connected layer that maps the extracted features into the task embedding dimension $d_e$. The base-learner is also a four-layer $3 \times 3$ convolutions with 32 filters, followed by a one-layer fully-connected layer that projects the output to prediction scores for 5 classes. Each convolution layer is accompanied with batch normalization, ReLU nonlinearity, and $2 \times 2$ max-pooling.

For all the compared methods, we follow the hyper-parameters settings for meta-training in MAML (Finn et al., 2017), which sets the adaptation learning rate $\alpha$ as 0.01, the meta-update learning rate $\beta$ as 0.001, the number of adaptation steps $\tau$ as 5, the meta batch size $|\mathcal{B}|$ as 4, the size of test set $|\mathcal{D}_{\mathcal{T}_i}^{te}|$ as 15, and the number of meta-update iterations as 60,000. For all the task-aware modulation methods (i.e., MMAML (Vuorio et al., 2019), L2F (Baik et al., 2020), HSML (Yao et al., 2019), ARML (Yao et al., 2020)), we set the dimension of task embedding $d_e$ as 128 (following Yao et al. (2019)). Additionally, for HSML (Yao et al., 2019), we set the number of clusters in the first and second layer of hierarchical tree as 4 and 2 respectively. For ARML (Yao et al., 2020), we set the number of vertices in the meta-knowledge graph as 4. For our proposed CTML, we set the number of clusters for both feature and path representations as 6, i.e., $k_{path} = k_{feat} = 6$.

## B.2  COLD-START RECOMMENDATION

Different from the image dataset, recommendation dataset typically consists of user-item interactions and profile features for both sides. The feature extractor is implemented as several embedding lookup tables, each corresponds to one categorical feature. We set the embedding dimension as 8 for all features. The base-learner is implemented as a general deep learning-based recommender system model, composed of embedding matrices followed by a two-layer fully connected layer that models the interaction between different features. The embedding dimension for base-learner is set to 8, and the hidden sizes for the fully-connected layers is set to [32, 16].

For all the compared methods, we follow the hyper-parameters settings for meta-training in MeLU (Lee et al., 2019), which sets the adaptation learning rate $\alpha$ as 5e-3, the meta-update learning rate $\beta$ as 5e-5, the number of adaptation steps $\tau$ as 5. For other hyper-parameters that are not explicitly specified by Lee et al. (2019), we set the meta batch size $|\mathcal{B}|$ as 1, the size of training set $|\mathcal{D}_{\mathcal{T}_i}^{tr}|$ for each user as 10 (the remaining are used as test set $\mathcal{D}_{\mathcal{T}_i}^{te}$ for that user, whose size varies for different users), and the number of meta-training epochs as 20. For MAMO (Dong et al., 2020) which also employs K-means clustering, we apply the same hyper-parameters tuning procedure as our CTML. That is, we tune both $k_{path}$ and $k_{feat}$ in $\{4, 8, 12, 16, 20\}$, and tune the dimension of task embedding $d_e$ in $\{64, 128, 256, 512, 1024\}$. Other hyper-parameters settings specific to the baseline methods are directly adopted from the source codes released by the authors.

## C  OTHER PATH LEARNER DESIGNS

In addition to the GRU-based meta path learner design described in equation 2, we also experiment on 3 other network designs to test their effectiveness in modeling the step-wise geometric quantities.

Recall that the meta path learner is applied coordinate-wise on the base-learner parameters. That is, the same meta path learner is used to process the sequence of step-wise tuples $\mathbf{P}_{\mathcal{T}_i}^d \in \mathbb{R}^{(\tau+1)\times 4}$ independently for all parameters at different dimensions $d \in \{1, 2, ..., D\}$.

The first design linearly transforms the vectorized $\mathbf{P}_{\mathcal{T}_i}^d$ with learnable weight and bias:

$$\texttt{Linear}: \quad \mathbf{h}_i^d = \mathbf{W} \cdot \text{vec}(\mathbf{P}_{\mathcal{T}_i}^d) + \mathbf{b}. \tag{10}$$

The second one employs multiple fully-connected layers with nonlinear activation $\sigma(\cdot)$:

$$
\begin{aligned}
\texttt{FC}: \quad \mathbf{u}^0 &= \text{vec}(\mathbf{P}_{\mathcal{T}_i}^d), \\
\mathbf{u}^1 &= \sigma(\mathbf{W}^1 \cdot \mathbf{u}^0 + \mathbf{b}^1), \\
&\vdots \\
\mathbf{u}^L &= \sigma(\mathbf{W}^L \cdot \mathbf{u}^{L-1} + \mathbf{b}^L), \\
\mathbf{h}_i^d &= \mathbf{u}^L.
\end{aligned}
\tag{11}
$$

The third design applies multiple self-attention layers on the sequence with scaled dot-product similarity following Vaswani et al. (2017). To inform about the sequential order, we apply sinusoid positional encodings at the input level, and also apply causal mask to prevent interaction with future steps (i.e., set the upper triangular entries of the similarity matrix to $-\infty$ before inputting into softmax). The final representation $\mathbf{h}_i^d$ is given by the $(\tau+1)$-th row (i.e., the last row) of the final-layer output:

$$
\begin{aligned}
\texttt{Attention}: \quad \mathbf{U}^0 &= \mathbf{P}_{\mathcal{T}_i}^d + positional\ encodings, \\
\mathbf{U}^1 &= \text{softmax}\left(\frac{\mathbf{Q}^1 \cdot (\mathbf{K}^1)^\top}{\sqrt{d_a}}\right) \cdot \mathbf{V}^1, \\
&\vdots \\
\mathbf{U}^L &= \text{softmax}\left(\frac{\mathbf{Q}^L \cdot (\mathbf{K}^L)^\top}{\sqrt{d_a}}\right) \cdot \mathbf{V}^L, \\
\mathbf{h}_i^d &= \mathbf{U}_{\tau+1,:}^L, \\
\text{where} \quad \mathbf{Q}^l &= \sigma(\mathbf{U}^{l-1} \cdot \mathbf{W}_Q^l + \mathbf{b}_Q^l) \in \mathbb{R}^{(\tau+1)\times d_a}, \\
\mathbf{K}^l &= \sigma(\mathbf{U}^{l-1} \cdot \mathbf{W}_K^l + \mathbf{b}_K^l) \in \mathbb{R}^{(\tau+1)\times d_a}, \\
\mathbf{V}^l &= \sigma(\mathbf{U}^{l-1} \cdot \mathbf{W}_V^l + \mathbf{b}_V^l) \in \mathbb{R}^{(\tau+1)\times d_a}.
\end{aligned}
\tag{12}
$$

Concatenating the path representations at all $D$ dimensions and passing it through a fully-connected layer gives the final path embedding $\mathbf{e}_{\mathcal{T}_i}^{path} \in \mathbb{R}^{d_e}$ for task $\mathcal{T}_i$:

$$\mathbf{e}_{\mathcal{T}_i}^{path} = \text{FC}([\mathbf{h}_i^1, ..., \mathbf{h}_i^D]^\top). \tag{13}$$

# D    ADDITIONAL EXPERIMENTAL RESULTS

## D.1    PERFORMANCE ON RESNET-12

We have conducted additional experiments to test the effectiveness of our method using a deeper network ResNet-12 as the backbone (i.e., the base-learner). We follow the implementation of Sun et al. (2019) which adopts (64-128-256-512) for the number of filters at the four blocks.

It was shown recently that, with a deeper backbone, simple training paradigm like pre-training and fine-tuning (Tian et al., 2020; Chen et al., 2018) can achieve very satisfactory performance. We therefore further include the following 3 baselines for comparison on ResNet-12:

- Finetune: This baseline simply pretrains a global feature extractor on the entire training dataset and then finetunes the classifier for individual tasks.

- Finetune-distance (Chen et al., 2018): This method modifies Finetune by computing cosine distance between the input feature vector and the class vectors for the classifier.

- Finetune-distill (Tian et al., 2020): This method modifies Finetune by employing a sequential self-distillation technique to pretrain the feature extractor.

Table 6 below presents the 5-way 1-shot performance on the Mixture-Of-Datasets with 2 different backbones: Conv-4 and ResNet-12. We can see that the gaps between Finetune methods and MAML are narrower for deeper backbone, which can be attributed to that the deeper models have larger capacity to accommodate the transferable knowledge obtained from the pre-training (in line with the findings of Chen et al. (2018)). However, under this task-heterogeneous setting, the task-conditioned methods still yield better performance than MAML and the Finetune baselines for ResNet-12, demonstrating the effectiveness of task-conditioning even for deeper backbone. And our method with path representation yields the best result.

Table 6: 5-way 1-shot performance (mean accuracy (%) $\pm$ std over 8 trials) on Mixture-Of-Datasets for Conv-4 and ResNet-12

| Methods | Mixture-Of-Datasets | |
|---|---|---|
| | Conv-4 | ResNet-12 |
| Finetune | $45.76 \pm 0.75$ | $53.48 \pm 0.57$ |
| Finetune-distance | $46.82 \pm 1.32$ | $54.67 \pm 1.44$ |
| Finetune-distill | $47.41 \pm 1.67$ | $55.72 \pm 1.07$ |
| MAML | $52.83 \pm 1.54$ | $56.84 \pm 1.38$ |
| MMAML | $54.28 \pm 1.30$ | $59.12 \pm 1.61$ |
| L2F | $56.61 \pm 1.21$ | $61.57 \pm 1.28$ |
| HSML | $56.32 \pm 1.16$ | $64.03 \pm 0.89$ |
| ARML | $57.21 \pm 1.68$ | $61.06 \pm 1.19$ |
| CTML | $\mathbf{59.03} \pm \mathbf{0.84}$ | $\mathbf{63.18} \pm \mathbf{1.02}$ |
| CTML(approx.) | $58.91 \pm 0.95$ | $62.69 \pm 1.32$ |

## D.2    PERFORMANCE ON MINIIMAGENET

Table 7 presents the results on MiniImagenet benchmark (Ravi & Larochelle, 2016), where tasks are constructed from a single dataset (i.e., there is only one underlying distribution) and hence can be considered as a task-homogeneous setting. Since in this case, task-aware methods do not have advantages over the global initialization methods, we see that all the baselines have comparable performance as the original MAML. Nevertheless, the results show that though the task-aware modulation is rendered less meaningful in task-homogeneous setting, CTML will yield performance at least comparable to the global initialization method in this scenario.

Table 7: Few-shot classification performance on MiniImagenet. Since for all the baselines on Mini-Imagenet, the results are reported in the form of 95% confidence interval of accuracy (%) based on 1000 meta-testing tasks, we also report our results in the same format for direct comparison.

| Methods | MiniImagenet | |
| --- | --- | --- |
| | 5-way 1-shot | 5-way 5-shot |
| MAML | $48.70 \pm 1.84$ | $63.11 \pm 0.90$ |
| MMAML | $49.86 \pm 1.85$ | $63.73 \pm 0.91$ |
| L2F* | $\mathbf{50.78} \pm \mathbf{1.72}$ | $64.06 \pm 0.89$ |
| HSML | $50.38 \pm 1.85$ | $64.03 \pm 0.90$ |
| ARML | $50.42 \pm 1.79$ | $64.12 \pm 0.90$ |
| CTML-feat | $50.29 \pm 1.82$ | $64.05 \pm 0.87$ |
| CTML-path | $50.43 \pm 1.82$ | $64.07 \pm 0.90$ |
| CTML | $50.47 \pm 1.83$ | $\mathbf{64.15} \pm \mathbf{0.90}$ |
| CTML(approx.) | $50.25 \pm 1.84$ | $64.03 \pm 0.93$ |

\* reproduced due to different implementation of base-learner.

## D.3 PERFORMANCE ON SUB-DATASETS

Table 8 presents the few-shot classification performance of CTML and the baselines on each sub-dataset. We see that CTML consistently outperforms all the baselines for all the sub-datasets. Other trends are similar to that of the overall results in Table 1a.

Table 8: Few-shot classification performance on individual sub-datasets. We sample 1000 tasks from each sub-dataset for meta-testing. The results are reported in the form of mean accuracy (%) $\pm$ std over 8 trials.

| | Methods | Bird | Aircraft | Fungi | Flower | Texture | Traffic Sign |
| --- | --- | --- | --- | --- | --- | --- | --- |
| | MAML | $52.73 \pm 1.52$ | $49.04 \pm 1.03$ | $39.42 \pm 1.25$ | $61.73 \pm 1.42$ | $31.80 \pm 1.12$ | $93.16 \pm 1.01$ |
| | MMAML | $54.02 \pm 1.36$ | $51.81 \pm 1.08$ | $41.99 \pm 1.27$ | $64.36 \pm 1.21$ | $32.83 \pm 1.54$ | $92.07 \pm 1.13$ |
| | L2F | $57.13 \pm 1.52$ | $54.23 \pm 1.14$ | $43.44 \pm 1.31$ | $64.79 \pm 1.47$ | $33.64 \pm 1.52$ | $92.61 \pm 0.87$ |
| | HSML | $55.75 \pm 1.14$ | $53.02 \pm 1.47$ | $43.03 \pm 1.24$ | $64.15 \pm 1.62$ | $34.07 \pm 1.48$ | $92.01 \pm 1.08$ |
| 5-way 1-shot | ARML | $58.12 \pm 1.53$ | $54.11 \pm 1.52$ | $44.15 \pm 1.12$ | $65.36 \pm 1.01$ | $33.87 \pm 1.39$ | $93.76 \pm 1.12$ |
| | CTML-feat | $58.72 \pm 1.33$ | $53.87 \pm 1.07$ | $44.21 \pm 1.37$ | $65.41 \pm 1.22$ | $33.07 \pm 1.51$ | $92.95 \pm 1.31$ |
| | CTML-path | $58.63 \pm 1.04$ | $54.21 \pm 1.37$ | $44.10 \pm 1.33$ | $66.72 \pm 1.37$ | $33.15 \pm 1.16$ | $94.09 \pm 0.92$ |
| | CTML | $\mathbf{61.05} \pm \mathbf{1.56}$ | $\mathbf{56.13} \pm \mathbf{1.21}$ | $\mathbf{45.19} \pm \mathbf{1.37}$ | $67.08 \pm 1.24$ | $\mathbf{35.24} \pm \mathbf{1.07}$ | $\mathbf{94.93} \pm \mathbf{0.89}$ |
| | CTML(approx.) | $60.73 \pm 1.42$ | $55.92 \pm 1.11$ | $44.72 \pm 1.61$ | $\mathbf{67.32} \pm \mathbf{1.05}$ | $34.98 \pm 1.26$ | $94.71 \pm 1.07$ |
| | MAML | $71.02 \pm 0.82$ | $63.64 \pm 0.43$ | $53.61 \pm 0.48$ | $75.29 \pm 0.55$ | $44.76 \pm 0.63$ | $97.01 \pm 0.18$ |
| | MMAML | $71.82 \pm 0.48$ | $67.01 \pm 0.54$ | $52.80 \pm 0.43$ | $77.63 \pm 0.84$ | $45.29 \pm 0.38$ | $98.13 \pm 0.57$ |
| | L2F | $72.64 \pm 0.71$ | $70.24 \pm 0.67$ | $56.79 \pm 0.68$ | $78.24 \pm 0.87$ | $45.91 \pm 0.76$ | $98.42 \pm 0.32$ |
| | HSML | $71.73 \pm 0.62$ | $71.26 \pm 0.39$ | $56.48 \pm 0.47$ | $79.12 \pm 0.49$ | $46.11 \pm 0.84$ | $98.22 \pm 0.33$ |
| 5-way 5-shot | ARML | $73.84 \pm 0.61$ | $72.07 \pm 0.15$ | $55.87 \pm 0.52$ | $79.67 \pm 0.44$ | $45.48 \pm 0.36$ | $98.52 \pm 0.26$ |
| | CTML-feat | $71.92 \pm 0.73$ | $72.19 \pm 0.45$ | $55.64 \pm 0.58$ | $78.94 \pm 0.82$ | $45.43 \pm 0.46$ | $98.43 \pm 0.37$ |
| | CTML-path | $73.65 \pm 0.29$ | $74.03 \pm 0.52$ | $56.78 \pm 0.62$ | $80.92 \pm 0.80$ | $46.18 \pm 0.33$ | $\mathbf{98.95} \pm \mathbf{0.51}$ |
| | CTML | $\mathbf{75.12} \pm \mathbf{0.27}$ | $\mathbf{74.09} \pm \mathbf{0.72}$ | $57.15 \pm 0.46$ | $\mathbf{81.07} \pm \mathbf{0.41}$ | $\mathbf{46.88} \pm \mathbf{0.56}$ | $98.81 \pm 0.18$ |
| | CTML(approx.) | $74.57 \pm 0.41$ | $73.51 \pm 0.41$ | $\mathbf{57.41} \pm \mathbf{0.51}$ | $80.71 \pm 0.71$ | $46.81 \pm 0.51$ | $98.72 \pm 0.31$ |

## D.4 PERFORMANCE ON RECOMMENDATION DATASETS

Table 9 presents the rating prediction performance on 3 datasets evaluated in two metrics: mean absolute error (MAE) and normalized discounted cumulative gain at top 20 (NDCG@20). The meta-testing users are further divided into warm and cold users, where warm users are those that have appeared in the meta-training set, and cold users are those that have not[11]. From the results, we see that CTML consistently outperforms the baselines.

---

[11]Note that for few-shot image classification, all the meta-testing tasks are unseen during meta-training, while for recommendation, it is possible that the meta-testing users also appeared in meta-training set.

Table 9: Rating prediction performance on 3 datasets in MAE and NDCG@20. Note that the results under 'All Users' are the same as that in Table 2.

| Datasets | Methods | MAE ↓ | | | NDCG@20 ↑ | | |
|---|---|---|---|---|---|---|---|
| | | All Users | Warm Users | Cold Users | All Users | Warm Users | Cold Users |
| MovieLens-1M | MeLU | 0.8164 | 0.8112 | 0.8197 | 0.7839 | 0.7935 | 0.7716 |
| | PAML | 0.7725 | 0.7575 | 0.7879 | 0.8255 | 0.8391 | 0.8165 |
| | MAMO | 0.8084 | 0.8073 | 0.8091 | 0.8118 | 0.8274 | 0.8030 |
| | MetaHIN | 0.7727 | 0.7490 | 0.7841 | 0.8264 | 0.8356 | 0.8110 |
| | CTML-feat | 0.7560 | 0.7482 | 0.7631 | 0.8470 | 0.8530 | 0.8384 |
| | CTML-path | 0.7592 | 0.7474 | 0.7690 | 0.8493 | 0.8499 | 0.8437 |
| | CTML | **0.7543** | 0.7415 | **0.7626** | **0.8562** | **0.8634** | **0.8487** |
| | CTML(approx.) | 0.7555 | **0.7411** | 0.7652 | 0.8531 | 0.8627 | 0.8418 |
| Yelp | MeLU | 0.9464 | 0.9100 | 0.9576 | 0.8149 | 0.8161 | 0.8139 |
| | PAML | 0.9280 | 0.9151 | 0.9336 | 0.8215 | 0.8230 | 0.8204 |
| | MAMO | 0.8965 | 0.9125 | 0.8840 | 0.8332 | 0.8296 | 0.8384 |
| | MetaHIN | 0.8832 | 0.9021 | 0.8767 | 0.8375 | 0.8324 | 0.8412 |
| | CTML-feat | 0.9023 | 0.9107 | 0.8949 | 0.8295 | 0.8301 | 0.8259 |
| | CTML-path | 0.8714 | 0.8912 | 0.8581 | 0.8401 | 0.8327 | 0.8420 |
| | CTML | **0.8603** | **0.8700** | **0.8533** | **0.8467** | **0.8364** | **0.8524** |
| | CTML(approx.) | 0.8679 | 0.8721 | 0.8607 | 0.8433 | 0.8319 | 0.8497 |
| Amazon-CDs | MeLU | 0.7216 | 0.6840 | 0.7383 | 0.9026 | 0.9037 | 0.9020 |
| | PAML | 0.7150 | 0.6813 | 0.7317 | 0.9083 | 0.9096 | 0.9051 |
| | MAMO | 0.7207 | 0.7103 | 0.7260 | 0.9116 | 0.9067 | 0.9140 |
| | MetaHIN | 0.6743 | 0.6675 | 0.6804 | 0.9111 | 0.9082 | 0.9153 |
| | CTML-feat | 0.6804 | 0.6697 | 0.6874 | 0.9122 | 0.9078 | 0.9179 |
| | CTML-path | 0.6286 | 0.6207 | 0.6372 | 0.9201 | 0.9180 | 0.9238 |
| | CTML | **0.6048** | **0.6080** | 0.6011 | **0.9279** | **0.9244** | **0.9293** |
| | CTML(approx.) | 0.6087 | 0.6144 | **0.6007** | 0.9244 | 0.9215 | 0.9260 |

### D.5 ADDITIONAL LEARNING PATH VISUALIZATION

#### D.5.1 FEW-SHOT IMAGE CLASSIFICATION

Here we provide more examples for visualizing the path learning results for the few-shot image classification experiments. Figure 5, 6 and 7 each shows the path learning results of 6 tasks randomly sampled from 3 of the 6 sub-datasets.

We see that in this application scenario, the overall direction of the learning path plays an important role in determining the task representation. Looking at the GRU gates at each step, it seems that the first 2 to 3 steps are often the most important/informative steps, after which the learning begins to converge and the subsequent steps no longer provide useful information about the learning direction.

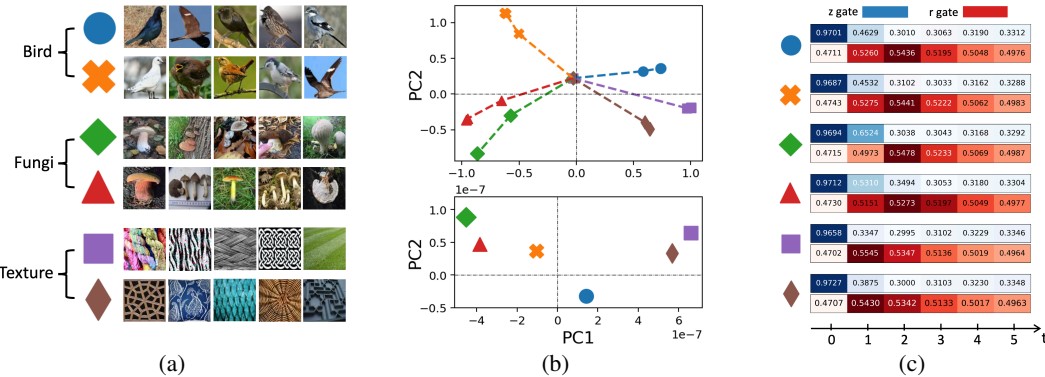

Figure 5: Visualizing path learning results of 6 tasks (5-way 1-shot) randomly sampled from Bird, Fungi and Texture sub-datasets (2 tasks from each sub-dataset).

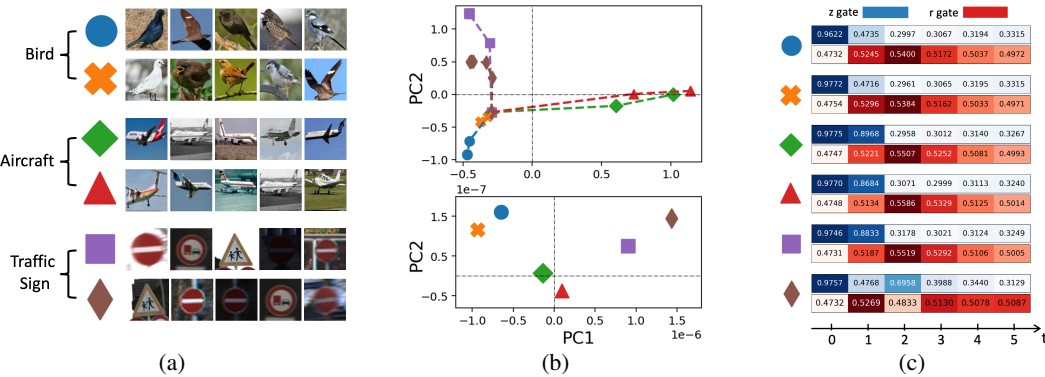

Figure 6: Visualizing path learning results of 6 tasks (5-way 1-shot) randomly sampled from Bird, Aircraft and Traffic Sign sub-datasets (2 tasks from each sub-dataset).

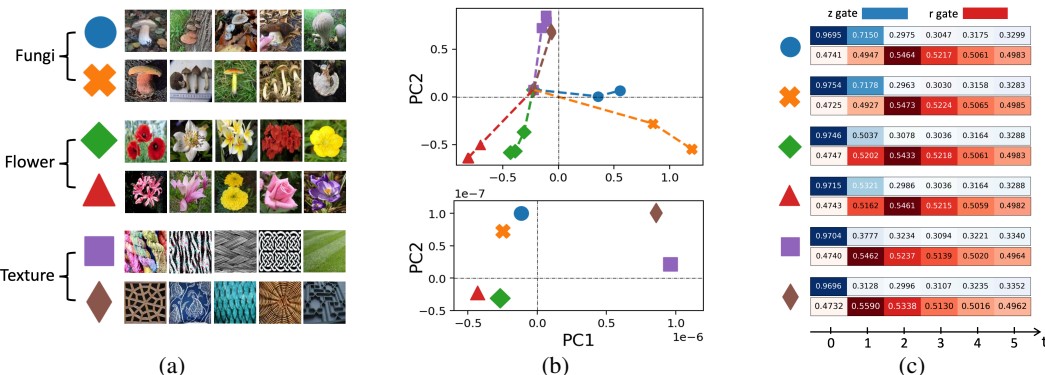

Figure 7: Visualizing path learning results of 6 tasks (5-way 1-shot) randomly sampled from Fungi, Flower and Texture sub-datasets (2 tasks from each sub-dataset).

### D.5.2 COLD-START RECOMMENDATION

We also provide examples for visualizing the learning paths for the cold-start recommendation experiments. Figure 8 and 9 each shows the path learning results of 6 users randomly selected from the MovieLens-1M dataset, and Figure 10 and 11 each shows the results of 6 users sampled from the Amazon-CDs dataset.

In this application scenario, we notice that apart from the overall direction of the path, the length of the path also plays an important role in determining the task representation. For instance, in Figure 11, the blue, orange, green and brown paths (upper plot) are clustered closely in the representation space (lower plot) due to similar lengths, although they are pointing at very different directions. Similar observation can also be found in Figure 8. Although blue and orange paths are pointing at the same directions, their embeddings are positioned far away from each other due to the big difference in path lengths.

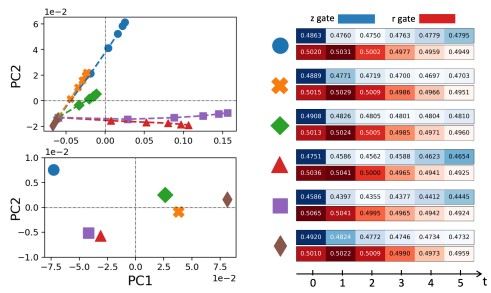

Figure 8: Visualizing path learning results of 6 users randomly selected from MovieLens-1M.

Figure 9: Visualizing path learning results of 6 users randomly selected from MovieLens-1M.

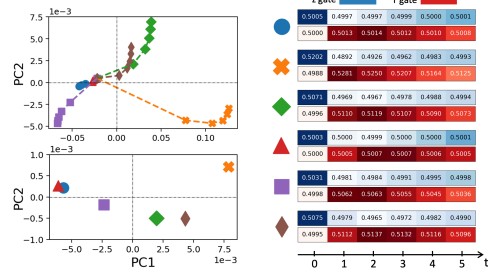

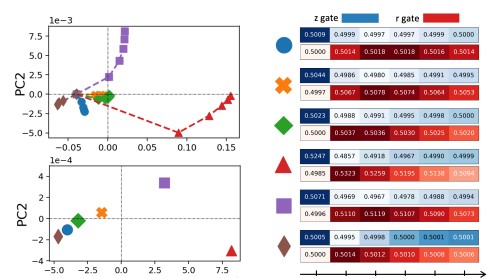

Figure 10: Visualizing path learning results of 6 users randomly selected from Amazon-CDs.

Figure 11: Visualizing path learning results of 6 users randomly selected from Amazon-CDs.

### D.6 ADDITIONAL SHORTCUT APPROXIMATION VISUALIZATION

In this section, we give more examples to verify the effectiveness of the shortcut approximation. Figure 12 shows the approximation results of 2 users sampled from the MovieLens-1M dataset, and Figure 13 shows the results of 3 tasks sampled from 3 different image sub-datasets. We can see that the shortcut tunnel is able to learn and accommodate different mappings, provide guarantee on the performance of the approximated model.

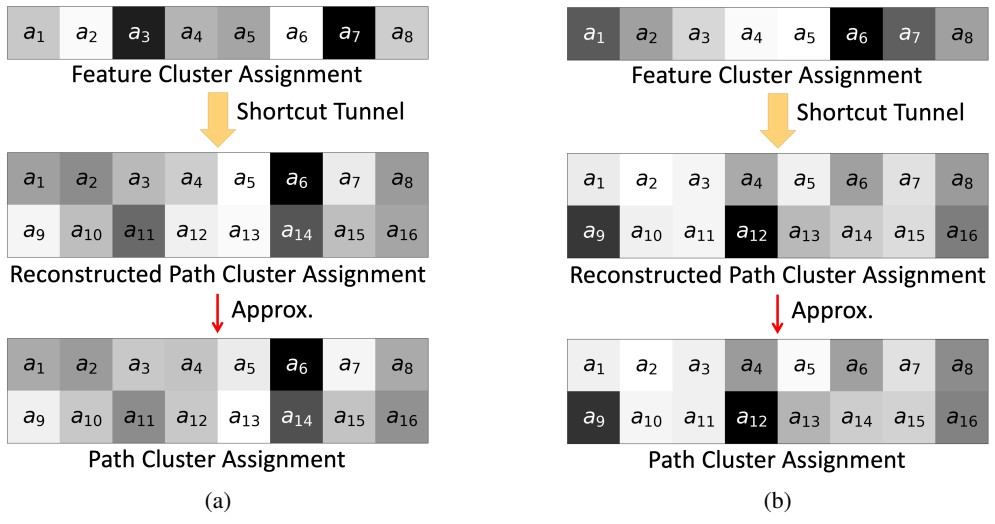

Figure 12: Shortcut approximation of path cluster assignment from feature cluster assignment for 2 users sampled from MovieLens-1M dataset. Here, we set $k_{feat} = 8$ and $k_{path} = 16$.

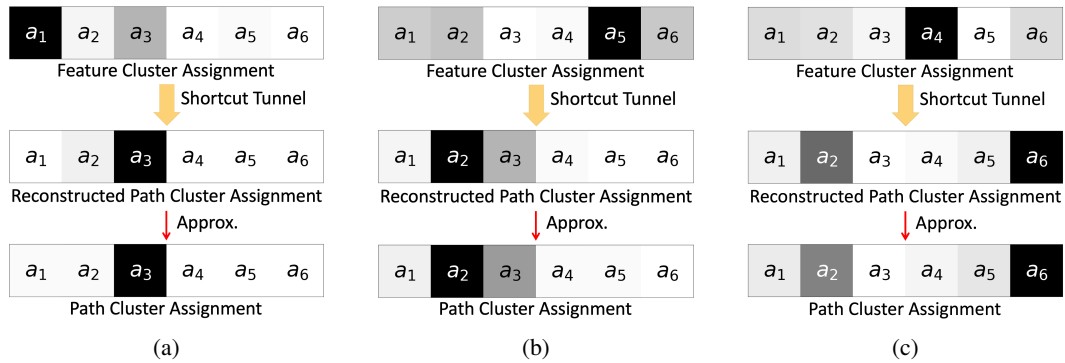

Figure 13: Shortcut approximation of path cluster assignment from feature cluster assignment for 3 tasks sampled from 3 different image sub-datasets. Here, we set $k_{feat} = k_{path} = 6$.

