# OpenReview forum: "Clustered Task-Aware Meta-Learning by Learning from Learning Paths"
_ICLR.cc/2022/Conference — ICLR 2022 Submitted_

### Official Review · Reviewer_kvxn · 2021-11-02

**Correctness:** 3
**Technical Novelty And Significance:** 2
**Empirical Novelty And Significance:** 2
**Recommendation:** 6
**Confidence:** 4

**Main Review:**

Strengths

Characterizing a task using the trajectory is an interesting perspective. The authors provide sufficient intuition and motivation behind the proposed characterization (the four entities in the tuple given as input to the GRU) of the task using trajectory information.

Though superficial, the analysis on trying to demystify the GRU’s learnings is interesting.

The paper is well written.

Concerns

The assumption behind the shortcut tunnel is that the tasks with similar feature assignments will also have identical path assignments. If that is the case, why does one have to rely on path characterization? Isn’t it going against the basic premise of the paper?

The improvements over the current state-of-the-art approaches on meta-dataset and the miniimagenet (in the appendix) are marginal at best. The goal of the meta-learning framework, such as the one used in the paper, is to learn a well-generalizable meta-model. Recent works have demonstrated significant improvements in the performances by using deep architectures. The gain on miniimagenet for both 5.1 and 5.5 settings is over 10% (Liu et al. 2020). Furthermore, it appears that CTML does not yield even the marginal gains when the number of shots increases (based on the results from the miniimagenet dataset). Thus, I am not convinced about the utility of such a complex method to squeeze out the last drop in performance when better models are readily available. It would be interesting to see the results of these deeper networks-based meta-learning models on the meta-dataset curated by the authors.


**Summary Of The Paper:**

The paper proposes to apply the task-aware-modulation on global Meta-learning for learning on tasks sampled from heterogeneous distributions. This paper builds on prior work on the feature-based task characterization to integrate the rehearsed task gradient descent trajectory into task representation. As the rehearsal task is computationally expensive, the authors learn a different network to estimate the rehearsed task-trajectory characterization from the feature representation. The proposed framework is tested on few-shot image classification (meta dataset and miniimagenet) and cold-start recommendation tasks.


**Summary Of The Review:**

Overall, it is a well-written paper with interesting and, at times, conflicting ideas. I deeply appreciate the amount of work invested by the authors. However, I am not convinced about the approach’s utility.

---post rebuttal update

Thank you for the clarification on the 'shortcut tunnel assumption. The assumption results in estimating the path cluster assignments through feature cluster assignments. This means that the information related to the path cluster assignments is already present (linear or non-linear relationship) in the feature cluster assignments. Thus, I am not entirely convinced that the model is getting any additional information to justify the improvement in the results because of the addition of the path cluster assignments.

The additional experiments using the heavier backbone strengthen the result.  I agree that on the curated 'meta-data set the proposed approach has an edge over prior methods. The authors have incorporated these results in the updated draft.

---

> ### Author Response · Authors · 2021-11-18
> **Response to Reviewer kvxn**
>
> Dear Reviewer:
>
> Thank you so much for your valuable comments! Regarding your concern, we have provided our responses as follows.
>
> **1. Assumption behind the 'shortcut tunnel'**
>
> We would like to apologize that the phrasing of our assumption behind the 'shortcut tunnel' is rather inaccurate. What we meant is that we assume there exists a one-to-one mapping between the feature cluster assignments and the path cluster assignments. That is, two tasks with the exact same feature assignments (i.e., the soft probability distribution over the feature clusters) will correspond to the same path assignment (i.e., the soft probability distribution over the path clusters). To justify this assumption, two tasks having the exact same feature assignments means that they have the same feature embeddings, which implies that these two tasks consist of exactly the same set of sample images. This set of images should correspond to only one set of labels (if the labeling is properly done), and hence resulting in only one possible learning path and path assignment.
>
> This assumption does not negate the main motivation of this paper, as the path assignment mapped from the feature assignment can be very different from the feature assignment (i.e., different probability distribution over the feature clusters and path clusters, and different dimensionality of the assignment vectors if the number of clusters is different for feature and path embeddings), and the positions of the cluster centroids in the respective latent space can also be very different for feature and path embeddings. In our experiments, we have demonstrated that adding path on top of feature serves to boost the performance (the comparison between CTML and CTML-feat in Table 1(a) and Table 2), which can be attributed to the additional information brought about by the path representation.
>
> Moreover, to rectify the claim we have initially made, tasks with similar feature assignments can actually have very different path assignments. Assuming two tasks with very similar feature embeddings (i.e., the visual features of the images that belong to the two tasks are very similar) but are actually drawn from different classes (i.e., the labels are different). In this case, the two tasks will be clustered closely in the space of feature embeddings (i.e., similar feature assignments), but positioned far apart in the space of path embeddings (i.e., different path assignments) as their learning paths will be very different. Generally, the mapping between the feature and path cluster assignments can be linear or non-linear, and we approximate it using a multilayer neural network.
>
> Lastly, we would like to thank you for pointing out this misleading sentence which does not reflect what we truly meant. We will revise this part accordingly.
>
> **3. Performance comparison with deeper backbone**
>
> As suggested in your review, we have conducted additional experiments to compare our method against the baselines using deeper backbone ResNet-12 on the Meta-Dataset created by us. The table below presents the 5-way 1-shot performance for two backbones: Conv-4 and ResNet-12.
>
> | Methods     | Conv-4 | ResNet-12 |
> | ----------- | ------ | --------- |
> | MAML        | 52.83  | 56.84     |
> | MMAML       | 54.28  | 59.12     |
> | L2F         | 56.61  | 60.69     |
> | HSML        | 56.32  | 61.57     |
> | ARML        | 57.21  | 61.06     |
> | CTML (ours) | 59.03  | 63.18     |
>
> From the results, we can see that with deeper backbone, our method still manages to outperform the baselines. Some may argue that the less complex meta-learning algorithms like MMAML and L2F, when applied on deeper backbone (ResNet-12), are able to outperform CTML on shallower backbone (Conv-4). However, we should also keep in mind that a shallower backbone has some important advantages over the deeper backbone: it allows for faster adaptation to new tasks at online phase, and it is more lightweight to be deployed on the edge device. Hence, we believe that our method is still advantageous if we fairly compare the methods using the same backbone.

---

> ### Author Response · Authors · 2021-11-27
> **Response to Reviewer kvxn**
>
> Dear Reviewer,
>
> Did our detailed responses address your concerns?

---

### Official Review · Reviewer_oNDW · 2021-11-02

**Correctness:** 3
**Technical Novelty And Significance:** 2
**Empirical Novelty And Significance:** 3
**Recommendation:** 5
**Confidence:** 4

**Main Review:**

My opinion is that the work has some significant positives, for example:

- The paper is well written, mostly clear besides some small details (see "Minor points" below)
- The authors report experiments on several benchmarks and several ablation studies.
- Visualization of the rehearsed learning paths (figure 3b) is nice.

However, there are some problems with the study that make it fall slightly short, in particular:

1) It is actually not true that this is the first meta-learning work doing clustering on multi-step optimization trajectories. The work of https://arxiv.org/abs/2103.04691 is conceptually very similar, they run gradient descent in the inner loop for a few steps and they cluster the resulting trajectories across tasks. There are several differences with the author's work, trajectories are not processed by a RNN, and the method is applied to a very different context (multi-language inferece), but is otherwise very similar conceptually.

2) I'm not entirely convinced that the 1% or 2% gains with respect to other methods reported in table 1 and table 2 are due to the superiority of the proposed architecture. Is the capacity (e.g. number of parameters) of the different models taken into account? Furthermore, some of the performances reported in table one falls within the standard deviation of other methods, so it is not clear whether differences are significant.

3) In section 4.4., while deriving the "shortcut tunnel", the authors justify their method by stating that "tasks with similar feature assignments will also have similar path assignments." If that is true, then the entire motivation of the study breaks, since the method is formulated under the assumption that path representations actually add significant information that is not avaiable in the feature representations.

Minor points:
- Why talking about Natural gradient in section 4.1.1? That seems completely out of context. Besides, it is not true that the Fisher is equivalent to the Hessian for cross entropy loss in general, that is true only for very special models.
- It is unclear to me why "learning path can be interpreted as encoding the predictive distribution"
- Along the entire paper it remains unclear what the authors mean by "lower order" and "higher order geometric quantities"
- There is actually a dataset that is called "Meta-dataset" (http://arxiv.org/abs/1903.03096), which is different from the one the authors define. It would be great to do some effort to avoid confusion between the two.


**Summary Of The Paper:**

This paper looks at the problem of meta-learning with heterogeneous tasks.
This is an interesting problem that is receiving increasing attention in the past two or three years.
Several previous work address this problem by clustering task representations and let the meta-learner exploit the information about task cluster.
The authors claim that, for the first time, they cluster tasks not only at the level of the input representation (features), but also at the level of the optimization trajectory in parameter space.


**Summary Of The Review:**

Given the above points 1, 2 and 3 I think the paper is marginally below the acceptance threshold

---

> ### Author Response · Authors · 2021-11-18
> **Response to Reviewer oNDW (1/5)**
>
> Dear Reviewer:
>
> Thank you so much for your valuable comments! Regarding your concerns, we have provided our responses as follows.
>
> **Major Concerns**
>
> ---
>
> **1. Differences from TreeMAML (Garcia, Jezabel R., et al. 2021)**
>
> First of all, we would like to thank you for pointing out this interesting work (which we will refer to as TreeMAML in the following discussion). However, we believe that there are some very fundamental differences between TreeMAML and our work in terms of both **methodology** and **motivation**.
>
> * Methodology
>
>   Though TreeMAML and our work both involve clustering of tasks based on gradients or learning trajectories, they differ in **where** the clustering outcome is applied, and **at what scale** the clustering is performed.
>
>   **a) The clustering outcome is applied at different stages of task adaptation to achieve different purposes**.
>
>   For our work, we first rehearse the entire learning trajectory and then compare the path embedding to global centroids. The final cluster-aware embedding is used to condition the **global initialization** to tailor to the specific task at hand. However, for TreeMAML, adaptations of all tasks proceed from the same global initialization, and the clustering occurs at **each adaptation step** to compute the next gradient step for each cluster (by averaging the gradients of all tasks belonging to that cluster), which in turn gives rise to the hierarchical tree structure across all the adaptation steps. In other words, TreeMAML modifies MAML by **changing the direction of each adaptation step**, while our work modifies MAML by **conditioning the global initialization**.
>
>   **b) The clustering is performed at different scales.**
>
>   For our work, the cluster centroids are meta-learned during the meta-training phase, hence the clustering is conducted at **global level** where all the meta-training tasks are taken into account. However, for TreeMAML, the clustering is performed on the fly at task adaptation phase, which means that only tasks at the current batch are taken into account. Hence, TreeMAML performs clustering only at **batch level**.
>
>   Furthermore, for TreeMAML, it is unclear how to perform inference for a single new task, since at least a batch of tasks will be needed in order to perform meaningful clustering at online task adaptation (or the gradients of meta-training tasks will need to be restored). On the contrary, our method allows more convenient cluster assignment for a new task by directly comparing to the well-trained cluster centroids.
>
> * Motivation
>
>   For TreeMAML, the motivation for gradient aggregation at each adaptation step, according to the paper, is to "increase the signal", since "if each task has scarce data, gradient updates for single tasks are noisy". Hence, the motivation of TreeMAML can be understood as to **produce more stable task adaptation with lower variance**.  However, for our work (and many other task-conditioned initialization methods, which we believe are more related to our work), we aim to **produce different initializations to handle task heterogeneity**.
>
> To the best of our knowledge, we believe that we are still the first to consider task representation based on learning path for the purpose of conditioning the global initialization. Nevertheless, we can see that there are some important links between TreeMAML and our method in that we both perform clustering on task adaptation trajectories. We will revise our paper to include this work and briefly talk about the differences.

---

> ### Author Response · Authors · 2021-11-18
> **Response to Reviewer oNDW (2/5)**
>
> **2. Significance of performance gains**
>
> Firstly, we would like to clarify that, instead of **"1% or 2%"**  gains (which you have pointed out in your review), the percentage gains over the baselines is at least **3%** for 5-way 1-shot case and **4%** for 5-way 5-shot case. And for cold-start recommendation, the percentage increase is at least **2%** for MovieLens and Yelp, and **10%** for Amazon. The following 2 tables record the percentage gains of CTML w.r.t. the baselines, corresponding to Table 1(a) and Table 2 in our paper:
>
> | Methods | 5-way 1-shot | 5-way 5-shot |
> | ------- | ------------ | ------------ |
> | MAML    | 11.74%       | 8.48%        |
> | MMAML   | 8.75%        | 6.43%        |
> | L2F     | 4.27%        | 5.45%        |
> | HSML    | 4.81%        | 5.02%        |
> | ARML    | 3.18%        | 4.17%        |
>
> | Methods | MovieLens | Yelp  | Amazon |
> | ------- | --------- | ----- | ------ |
> | MeLU    | 7.61%     | 9.10% | 16.19% |
> | PAML    | 2.36%     | 7.30% | 15.41% |
> | MAMO    | 6.69%     | 4.04% | 16.08% |
> | MetaHIN | 2.38%     | 2.59% | 10.31% |
>
> As suggested in your review, we record the parameter size of our method and the baselines in the table below.
>
> | Methods     | Parameter Size |
> | ----------- | -------------- |
> | MMAML       | 4,375,146      |
> | L2F         | 7,196,773      |
> | HSML        | 14,171,889     |
> | ARML        | 14,369,394     |
> | CTML (ours) | 8,555,799      |
>
> We can see that our method is less parameterized than HSML and ARML, but is able to outperform them with around 3 ~ 5% performance gains (according to the table above). As compared to L2F, our method has slightly larger parameter size but allows for 4 ~ 5% performance gains. And for MMAML, our method has twice the size of MMAML but with larger performance gains of 6 ~ 8%.
>
> Furthermore, to test the significance of the performance gain statistically, we've conducted $t$-test [1] based on the results of 8 trials of each method. The table below presents the $t$-value and degrees of freedom (df) computed based on the performance (5-way 1-shot) of CTML and the corresponding baseline, and the significance level at which the improvement of CTML over the baseline is considered statistically significant.
>
> | Methods | $t$-value | df    | Significance Level |
> | ------- | --------- | ----- | ------------------ |
> | MAML    | 7.69      | 13.89 | 0.0005             |
> | MMAML   | 6.32      | 13.17 | 0.0005             |
> | L2F     | 3.31      | 12.72 | 0.005              |
> | HSML    | 3.75      | 12.43 | 0.005              |
> | ARML    | 2.74      | 10.30 | 0.025              |
>
> We can see that at the worst case (vs ARML), the significance level is 0.025, which means that there is at least 97.5% confidence when we conclude that there is a performance gain over ARML. For the 5-way 5-shot case and the cold-start recommendation experiments, the significance levels are all below 0.0005. Therefore, we believe that the performance gains of our method can be considered quite significant.
>
> [1] We follow the unequal variances $t$-test, where $t\text{-value}=\frac{mean_1-mean_2}{\sqrt{\frac{var_1}{n_1}+\frac{var_2}{n_2}}}$ and $\text{df}=\frac{\left(\frac{var_1}{n_1}+\frac{var_2}{n_2}\right)^2}{\frac{\left(\frac{var_1}{n_1}\right)^2}{n_1-1}+\frac{\left(\frac{var_2}{n_2}\right)^2}{n_2-1}}$, and report the significance level based on the [$t$-table](https://www.sjsu.edu/faculty/gerstman/StatPrimer/t-table.pdf) for one-tailed test. As a side note, the significance level indicates the risk of concluding that there is a performance gain when in fact there isn't. The lower the significance level, the more significant the performance gain is.

---

> ### Author Response · Authors · 2021-11-18
> **Response to Reviewer oNDW (3/5)**
>
> **3. Assumption behind the 'shortcut tunnel'**
>
> We would like to apologize that the phrasing of our assumption behind the 'shortcut tunnel' is rather inaccurate. What we meant is that we assume there exists a one-to-one mapping between the feature cluster assignments and the path cluster assignments. That is, two tasks with the exact same feature assignments (i.e., the soft probability distribution over the feature clusters) will correspond to the same path assignment (i.e., the soft probability distribution over the path clusters). To justify this assumption, two tasks having the exact same feature assignments means that they have the same feature embeddings, which implies that these two tasks consist of exactly the same set of sample images. This set of images should correspond to only one set of labels (if the labeling is properly done), and hence resulting in only one possible learning path and path assignment.
>
> This assumption does not negate the main motivation of this paper, as the path assignment mapped from the feature assignment can be very different from the feature assignment (i.e., different probability distribution over the feature clusters and path clusters, and different dimensionality of the assignment vectors if the number of clusters is different for feature and path embeddings), and the positions of the cluster centroids in the respective latent space can also be very different for feature and path embeddings. In our experiments, we have demonstrated that adding path on top of feature serves to boost the performance (the comparison between CTML and CTML-feat in Table 1(a) and Table 2), which can be attributed to the additional information brought about by the path representation.
>
> Moreover, to rectify the claim we have initially made, tasks with similar feature assignments can actually have very different path assignments. Assuming two tasks with very similar feature embeddings (i.e., the visual features of the images that belong to the two tasks are very similar) but are actually drawn from different classes (i.e., the labels are different). In this case, the two tasks will be clustered closely in the space of feature embeddings (i.e., similar feature assignments), but positioned far apart in the space of path embeddings (i.e., different path assignments) as their learning paths will be very different. Generally, the mapping between the feature and path cluster assignments can be linear or non-linear, and we approximate it using a multilayer neural network.
>
> Lastly, we would like to thank you for pointing out this misleading sentence which does not reflect what we truly meant. We will revise this part accordingly.

---

> ### Author Response · Authors · 2021-11-18
> **Response to Reviewer oNDW (4/5)**
>
> **Minor Concerns**
>
> ---
>
> **1. Why talk about natural gradient? Fisher = Hessian?**
>
> Firstly, regarding the natural gradient descent, the reason we mentioned it here is just to provide some background to justify that the gradient descent described in equation 1 is the steepest (most efficient) descent direction of loss when the parameter space is Euclidean. According to [1], the natural gradient descent is defined as $-\tilde{\triangledown}_\theta\mathcal{L}=-G^{-1}\triangledown_\theta\mathcal{L}$, which is the steepest descent direction of $\mathcal{L}$ in a Riemannian space, and $G$ is the metric tensor of the Riemannian space. In particular, for the Euclidean space, the metric tensor $G$ is an identity matrix $I$, which gives $-\tilde{\triangledown}_\theta\mathcal{L}=-I^{-1}\triangledown_\theta\mathcal{L}=-\triangledown_\theta\mathcal{L}$ (the steepest descent direction is just the ordinary gradient descent). We figure that this part might be of interest as we have described the task manifold on the Riemannian space, but we realize that removing it will not affect understanding of the main idea being put forward in this paper. Hence, we will remove it for better clarity.
>
> Secondly, regarding the connection between the Fisher matrix and the Hessian matrix, we believe that the sentence is largely true based on our understanding from the literature. According to [2], the Fisher information matrix $F(\theta)$ of parameters $\theta$ is the negative expectation of the second derivative of log-likelihood $l(\theta)$ w.r.t. $\theta$, i.e., $F(\theta)=-E[\frac{\partial^2}{\partial{\theta^2}}l(\theta)]$. If the loss to be minimized is the *negative* log-likelihood  (which is just another name of the cross entropy loss for multiclass classiﬁcation problem [3]), the Hessian of the loss is equivalent to the Fisher matrix in expectation. We realize that our understanding may not be entirely correct. We will appreciate it if you can provide more insights on this.
>
> References:
>
> [1] Amari, Shun-Ichi. "Natural gradient works efficiently in learning." 1998.
>
> [2] Lehmann, Erich L., and George Casella. *Theory of point estimation*. 2006. (p.116, Lemma 5.3, eq. 5.16).
>
> [3] Anzai, Yuichiro. *Pattern recognition and machine learning*. 2012. (p. 209, eq. 4.108).

---

> > ### Comment · Reviewer_oNDW · 2021-11-28
> > **Response**
> >
> > The Fisher information matrix is not equal to the Hessian matrix even when the loss is equal to the negative log likelihood. The reason is that the Fisher information matrix is defined as an expectation over the model distribution, not the data distribution. This makes a huge difference. For example, it implies that the Fisher information matrix is positive semidefinite, while the Hessian matrix obviously is not positive semidefinite, otherwise neural networks optimization would be a convex problem!!!
> > Similarly, the Hessian cannot be used as a Riemann metric tensor because it may have negative eigenvalues.
> > Using the inverse Hessian instead of the inverse Fisher as a pre-conditioner implies that you are using Newton's method for optimization, instead of natural gradient descent. Newton's method does not work for optimizing neural networks, because the problem is non-convex and  it does not even converge!
> > Furthermore, Natural gradient descent is invariant for any smooth and invertible transformation (including non-linear), while Newton's method is not (it is invariant for affine transformations though).

---

> > > ### Author Response · Authors · 2021-11-29
> > > **Response to Reviewer oNDW**
> > >
> > > Dear Reviewer,
> > >
> > > Thank you so much for the reply!
> > >
> > > First of all, we would like to sincerely apologize for mixing up the concepts of the Fisher information matrix and the empirical Fisher approximation. You are absolutely right that the Fisher information matrix is defined as expectation over the parametric density function $p(x|\theta)$ instead of the empirical data distribution, and only with that the following equality holds: $F(\theta)=E_{p(x|\theta)}[(\frac{\partial \log{p(x|\theta)}}{\partial\theta})(\frac{\partial \log{p(x|\theta)}}{\partial\theta})^T]=-E_{p(x|\theta)}[\frac{\partial^2\log{p(x|\theta)}}{\partial^2\theta}]$. However, it is often challenging to compute the expectation of a complicated density function. Applying the Monte-Carlo approach is also expensive due to the cost of sampling the density estimate. Hence, we adopted the common approach to approximate the expectation with empirical Fisher using the training samples [1, 2]. That is, for a task $\mathcal{T}_i$, we compute the empirical Fisher $\hat{F}_i(\theta)=\frac{1}{|S_i|}\sum_\{x\in S_i\}(\frac{\partial \log{p(x|\theta)}}{\partial\theta})(\frac{\partial \log{p(x|\theta)}}{\partial\theta})^T$, where $S_i$ is the training/support set of task $\mathcal{T}_i$ (we only take the diagonal entries of the empirical Fisher assuming negligible correlation among different parameters). We use this quantity as an approximation to capture the second-order information and avoid the high cost of computing the exact Fisher information matrix. We have revised the corresponding part in the paper to correct the mistake in the Fisher information matrix definition and further explain that in practice we used the empirical Fisher.
> > >
> > > Note that one can also compute the Hessian matrix to represent the second-order information, but it is also computationally very slow.
> > >
> > > Empirically, we found that though the empirical Fisher is just an approximation of the second-order information, adding it as a feature is still beneficial for the path representation. In Table 1(b) in our paper, removing the empirical Fisher from the step-wise tuples results in some performance drops (-0.9 for 1-shot and -0.3 for 5-shot).
> > >
> > > Again, we would like to apologize for our mistake and express our sincerest gratitude for you pointing this out. We have revised the paper to reflect the correct definition of the Fisher information matrix and mentioned that in practice we used the empirical Fisher as an approximation. Please do let us know if you have other concerns.
> > >
> > > [1] Bottou, Léon, Frank E. Curtis, and Jorge Nocedal. "Optimization methods for large-scale machine learning." *Siam Review* 60.2 (2018): 223-311.
> > >
> > > [2] Park, Hyeyoung, S-I. Amari, and Kenji Fukumizu. "Adaptive natural gradient learning algorithms for various stochastic models." *Neural Networks* 13.7 (2000): 755-764.

---

> ### Author Response · Authors · 2021-11-20
> **Response to Reviewer oNDW (5/5)**
>
> **2. Why "learning path can be interpreted as encoding the predictive distribution"?**
>
> Note that the learning path of a task is directed by the labels (i.e., learning proceeds in direction that optimizes the loss). Hence, unlike the feature embeddings, which if plotted for all the tasks in the latent space will form the marginal distribution $p(\mathbf{x})$, the path embeddings in the latent space will constitute the conditional distribution $p(y|\mathbf{x})$.
>
> To illustrate our point, here we give a simple example on binary classification problem. Suppose each data instance $i$ consists of a 2-D feature $\mathbf{x}^{(i)}=[x_1, x_2]$ and a label $y^{(i)}\in\\{0,1\\}$. We apply the logistic model $\hat{y}=\sigma(z)=\frac{1}{1+e^{-z}}$,  where $z=\beta_0+\beta_1x_1+\beta_2x_2$, and minimize the log loss $\mathcal{l}=-y\log(\hat{y})+(1-y)\log(\hat{y})$. To further simplify the case, we assume each task consists of only one instance (that is, each instance is treated as a task). Consider 3 tasks $\\{\mathcal{T}_1, \mathcal{T}_2, \mathcal{T}_3\\}$ with exactly the same feature $[x_1=2, x_2=4]$ but different label $y$, we can compute the derivatives of $l$ w.r.t. $\beta_0$, $\beta_1$ and $\beta_2$ as follows:
>
> | $\mathcal{T}_i$ | $x_1$ | $x_2$ | $y$  | $\frac{\partial{l}}{\partial{\beta_0}}$ | $\frac{\partial{l}}{\partial{\beta_1}}$ | $\frac{\partial{l}}{\partial{\beta_2}}$ |
> | --------------- | ----- | ----- | ---- | --------------------------------------- | --------------------------------------- | --------------------------------------- |
> | 1               | 2     | 4     | 0    | $\frac{1}{e^z+1}$                       | $\frac{2}{e^z+1}$                       | $\frac{4}{e^z+1}$                       |
> | 2               | 2     | 4     | 0    | $\frac{1}{e^z+1}$                       | $\frac{2}{e^z+1}$                       | $\frac{4}{e^z+1}$                       |
> | 3               | 2     | 4     | 1    | $\frac{-1}{e^z+1}$                      | $\frac{-2}{e^z+1}$                      | $\frac{-4}{e^z+1}$                      |
>
> Since the learning path follows the gradient descent direction, we can see that $\mathcal{T}_1$ and $\mathcal{T}_2$ will have exactly the same learning paths, while $\mathcal{T}_3$ will have a learning path that points in opposite direction. Hence, the path embeddings generated for $\mathcal{T}_1$ and $\mathcal{T}_2$ will be positioned together in the latent space, while $\mathcal{T}_3$ will be positioned far apart. The overall 3-data point distribution in the latent space essentially describes the conditional distribution where $p(y=0|x_1=2, x_2=4)=\frac{2}{3}$ and $p(y=1|x_1=2, x_2=4)=\frac{1}{3}$. On the other hand, the feature embeddings which are not informed by the labels will position all the 3 tasks together, representing the marginal distribution $p(x_1=2, x_2=4)=1$.
>
> **3. "Lower-order" and "higher-order geometric quantities"?**
>
> Geometric quantities generally involve measurements like length, slope and curvature. By "higher-order geometric quantities", we are referring to the higher-order derivatives of the loss, like the second-order derivative which indicates the curvature of the loss surface. And by "lower-order geometric quantities", we are referring to the first-order derivative (i.e., the slope) and the zero-order derivative (i.e., the loss itself). However, we do realize that the phrasing used here is rather ambiguous and can cause confusion.  A better way of describing it will be like "higher/lower-order behaviors of the learning trajectory near the point at various adaptation steps".
>
> **4. Avoid confusion with the Meta-Dataset benchmark**
>
> About the naming of the datasets we used in our experiments, we will rename it to "Mixture-Of-Datasets" or "Multi-Datasets" to avoid confusion with the Meta-Dataset benchmark (Triantafillou et al. 2020).

---

> ### Author Response · Authors · 2021-11-27
> **Response to Reviewer oNDW**
>
> Dear Reviewer,
>
> Did our detailed responses address your concerns?

---

### Official Review · Reviewer_9y7H · 2021-11-02

**Correctness:** 3
**Technical Novelty And Significance:** 3
**Empirical Novelty And Significance:** 2
**Recommendation:** 6
**Confidence:** 4

**Main Review:**

Utilizing feature embeddings jointly with path embeddings drawn from several steps of the inner-loop optimization is novel to the best of my knowledge, and an interesting direction. The authors do a good job of motivating the need to consider geometric information from more than just the initialization point. The paper is also nicely written, well organized and for the most part easy to follow. However, I have a few concerns, primarily relating to the placement of this work into the previous literature. More details below.

- Limited comparisons with state-of-the-art methods. The experimental comparisons are only against similar MAML-based meta-learning methods. These are not state-of-the-art for few-shot classification, so considering only these in the table (though indeed they are most relevant to the proposed method) is misleading. In fact, it was recently shown that transfer learning approaches can match or surpass the performance of meta-learning methods [4,5], especially when using larger architectures than a 4-layer-convnet, while also having the advantage of being simpler and more computationally efficient. On mini-ImageNet, according to the results presented in the Appendix, there is a large gap between the proposed approach and recent work e.g. [4], which isn’t included in that table. Despite the nice ideas presented in this paper, the significance of the work is low if the results aren’t comparable with conceptually simpler and computationally more efficient baselines.

Relationship to related work:
- [1,2] should also be cited in the context of task-conditioned meta-learners (non-MAML-like in this case)
- It is unfortunate that the authors construct a new collection of datasets and name it Meta-Dataset, since Meta-Dataset is already the name of a few-shot classification benchmark [3] which actually has high overlap with the datasets chosen here. In fact, Meta-Dataset [3] is a good evaluation benchmark for approaches that deal with task heterogeneity like the one proposed here.
- “Though task-specific methods have been developed to tailor the initialization, they overlook generalization among similar tasks” - I would argue that this isn’t necessarily the case. Approaches that use a network to predict a task encoding, and then condition on that encoding (like Oreshkin et al, and [1], [2], etc) *do* consider relationships between tasks implicitly, since similar tasks are likely to cluster close to each other in the learned task embedding space, even if an explicit clustering objective isn’t employed. I would revise this sentence to more fairly describe previous work.

Scalability considerations
- The architecture considered for few-shot learning is a small 4-layer convnet. Is this method easy to scale up to larger ones like the resnet-12 and resnet-18 that are commonly used more recently [3,4,5]?
- How many steps of gradient descent were performed in these experiments? It seems that this method would not scale well to large numbers of steps which might be more appropriate for larger support set sizes? The evaluation here is constrained to 1-shot and 5-shot tasks.

Additional ablation
- It would be good to also ablate the effect of the clustering, for both the path and feature embeddings.

Less important comments and typos
- ‘The amount of information contained in the task-specific data about a “probe” network can be seen as a good representation of the task itself’. It should be explained what a “probe” network is
- ‘while the interaction between task data and the base-learner (e.g., gradients) is often not neglected.’ – I think you meant “is often neglected” here (without the not).
- ‘However, these methods utilize gradients only at a single point in parameter space’ - I would say ‘only at the initialization’, to make it clearer what the single point is, compared to what is proposed in this work.
- ‘we first elaborate task representation learning’ - the word ‘on’ missing after ‘elaborate’

Suggestion: I encourage the authors to consider the methodology of [1], which meta-learns the task-conditioning networks that are used to predict task-specific parameters for a pre-trained (and now frozen) feature extractor. This combination of pre-training and meta-learning is very effective and competitive with state-of-the-art in Meta-Dataset. Can some of the ideas presented here be used in the context of a meta-learning phase on a pre-trained feature extractor too? In contrast to the proposed method of this work, [1] only uses feature embeddings and doesn’t consider path embeddings (in fact there is no gradient descent adaptation in that work, it’s an amortized model).

References
- [1] Fast and Flexible Multi-Task Classification Using Conditional Neural Adaptive Processes. Requeima et al. NeurIPS 2019.
- [2] Improved Few-shot Visual Classification. Bateni et al. CVPR 2020.
- [3] Meta-Dataset: A Dataset of Datasets for Learning to Learn from Few Examples. Triantafillou et al. ICLR 2020.
- [4] Rethinking Few-Shot Image Classification: a Good Embedding Is All You Need? Tian et al.
- [5] A Closer Look at Few-shot Classification. Chen et al. ICLR 2019.


**Summary Of The Paper:**

This paper proposes a meta-learning approach based on MAML but with a task-conditioned initialization, which takes into account both information extracted from the features of the task (support set) data, as well as geometric information from a “rehearsed” learning path (obtained by gradient descent on the support set). The geometric information includes the parameters, loss, gradient and fisher matrix for different steps of the task adaptation process. They propose a GRU architecture to process this information across the different steps, yielding a path embedding. For both the feature and path embeddings, they utilize clustering, motivated by the need to share knowledge across similar tasks. They then combine the path and feature embeddings via an additional neural network to generate the final task-specific initialization. Computing the path embedding requires rehearsing which is computationally expensive (it’s akin to training twice on the task, once for rehearsing, and then training from the task-conditioned initialization). To amend this at inference time, they meta-learn a ‘tunnel’ connection that predicts the path embedding from the feature embedding. At test time, only a forward pass through this meta-learned connection is required instead of rehearsing. They experimentally evaluate their approach on few-shot classification and cold-start recommender problems and show performance gains over similar MAML-based approaches. They also perform ablation studies and analyses to understand the contribution of different parts of this system to downstream performance.

**Summary Of The Review:**

Overall, while the paper presents some nice ideas, is technically novel, well-organized and easy to follow, I have some concerns primarily about the relationship to prior work that make me lean towards rejection at this stage.

########
Edit after rebuttal:
During the discussion with the authors, they have addressed some of my concerns by running additional ablations (removing the clustering component), comparisons to transfer learning methods, and experimenting with deeper backbones. They also have commented on the scalability of their approach, and discussed pros and cons w.r.t efficiency at training time vs deployment for transfer learning vs meta-learning methods. Overall, my view is that test-time efficiency is important for certain applications, which makes meta-learning methods a good candidate, and the particular variant proposed here seems especially suited at the important problem of handling heterogeneous data and outperforms previous MAML-based meta-learning methods that were designed for this. Based on these, I updated my score from a 5 to a 6 to reflect that the paper is slightly above the publication bar in my opinion.

To strengthen the work further, I recommend comparing against recent methods developed for heterogeneous data on common benchmarks. Some examples are the following:
- Fast and Flexible Multi-Task Classification Using Conditional Neural Adaptive Processes. Requeima et al. NeurIPS 2019.
- Selecting Relevant Features from a Multi-domain Representation for Few-shot Classification. Dvornik et al. ECCV 2020.
- A Universal Representation Transformer Layer for Few-Shot Image Classification. Liu et al. ICLR 2021.
- Learning a Universal Template for Few-shot Dataset Generalization. Triantafillou et al. ICML 2021.
- Universal Representation Learning from Multiple Domains for Few-shot Classification. Li et al.
- Memory Efficient Meta-Learning with Large Images. Bronskill et al. NeurIPS 2021.

---

> ### Author Response · Authors · 2021-11-18
> **Response to Reviewer 9y7H (1/3)**
>
> Dear Reviewer:
>
> Thank you so much for your valuable comments! Regarding your concerns, we have provided our responses as follows.
>
> **1. Comparison with non-MAML baselines & deeper backbone**
>
> As suggested in your review, we have conducted additional experiments to compare our method against the state-of-the-art non-MAML methods (i.e., the two works mentioned in your review) using deeper backbone ResNet-12.
>
> We further include the following 3 baselines:
>
> (1) Finetune, which pre-trains a global feature extractor on the entire training dataset and then finetunes the classifier for individual tasks.
>
> (2) Finetune-cosine distance (Chen et al. 2019), which modifies Finetune by computing cosine distance between the input feature vector and the class vectors for the classifier.
>
> (3) Finetune-distill (Tian et al. 2020), which modifies *Finetune* by employing a sequential self-distillation technique to pre-train the global feature extractor.
>
> The table below presents the 5-way 1-shot performance on Meta-Dataset (the one created by us) for 2 backbones: Conv-4 and ResNet-12.
>
> | Methods                                     | Conv-4 | ResNet-12 |
> | ------------------------------------------- | ------ | --------- |
> | Finetune                                    | 45.76  | 53.48     |
> | Finetune-cosine distance (Chen et al. 2019) | 46.82  | 54.67     |
> | Finetune-distill (Tian et al. 2020)         | 47.41  | 55.72     |
> | MAML                                        | 52.83  | 56.84     |
> | MMAML                                       | 54.28  | 59.12     |
> | L2F                                         | 56.61  | 60.69     |
> | HSML                                        | 56.32  | 61.57     |
> | ARML                                        | 57.21  | 61.06     |
> | CTML (ours)                                 | 59.03  | 63.18     |
>
> We can see that the gaps between Finetune methods and MAML are narrower for deeper backbone, which is in line with the findings of Chen et al. (2019). This can be attributed to that the deeper model has larger capacity to accommodate the transferable knowledge obtained from the pre-training. Under this task-heterogeneous setting, the task-conditioned methods still yield better performance than MAML and the Finetune baselines for ResNet-12, demonstrating the effectiveness of task-conditioning even for deeper backbone. And our method with path representation yields the best result.
>
> Due to time constraints, we did not conduct further experiments on deeper backbone, like ResNet-34. But drawing from the experimental results of Chen et al. (2019) (Figure 3), further increasing the depth of ResNet only yields marginal improvements.
>
> Despite the significant gains in performance for the simple Finetune methods by simply using deeper backbone, we would like to highlight that there are still some important merits of MAML-based methods. Firstly, MAML-based methods explicitly optimize for faster adaptation for new tasks. That means from the learned initialization, we can effectively adapt to the new task with just few steps. However, for the Finetune methods, we usually require much more steps to attain convergence when finetuning for a new task. Secondly, MAML-based methods (especially the task-conditioned ones) can yield very satisfactory performance with shallower backbone, while Finetune methods will need a much deeper backbone to achieve comparable performance. A shallower backbone is always more desirable at the online deployment phase. It is faster to train when new task is coming, and more lightweight to be deployed on the edge device. Hence, despite the more complex algorithm and longer meta-training time of the MAML-based methods, they enjoy some benefits (i.e., faster adaptation, lightweight backbone) that the simple Finetune methods cannot achieve.
>
> **2. Relationship to related work**
>
> - Thank you for pointing out these related works. We will add them to the discussion of task-conditioned meta-learning methods.
> - About the naming of the datasets we used in our experiments, we will rename it to "Mixture-Of-Datasets" or "Multi-Datasets" to avoid confusion with the Meta-Dataset benchmark (Triantafillou et al. 2020).
> - We acknowledge that this sentence may not fairly describe the previous task-specific methods. We will revise it to "Though task-speciﬁc methods have been developed to tailor the global initialization, they lack explicit modeling of the global clustering structure". Hope this one will be better.

---

> > ### Comment · Reviewer_9y7H · 2021-11-20
> > **Response to author rebuttal**
> >
> > Thank you for the interesting discussion and additional experiments, including the ablation of removing the clustering, comparisons to other non-MAML based models and deeper backbones. A few additional questions below.
> >
> > - It’s strange that the proposed CTML outperforms e.g. Finetune-distill (by a large margin!) on the dataset used here, while the opposite is true (by a large margin again!) on mini-ImageNet, based on the results reported for CTML on mini-ImageNet in the Appendix. In fact, it’s strange that vanilla MAML also outperforms Finetune-distill here while again the opposite is reported to be true in (Tian et al, 2020). Specifically, Tian et al (2020) report that MAML’s accuracy is 48.7 / 63.1 % for 1 / 5 shots respectively, compared to 64.8 / 82.1% for Finetune-distill on mini-ImageNet (and according to the Appendix, CTML’s performance is 50.5 / 64.1% on mini-ImageNet). I understand that the proposed method shines in heterogeneous settings, but this doesn’t explain why even vanilla MAML works better than Finetune-distill in the results reported here. Do you have any thoughts on why this is?
> >
> > - The rebuttal mentions that, for larger support set sizes, the number of inner loop steps can be still kept small and MAML-like methods are ‘still being able to find a good initialization.’ I’m not really convinced that MAML-like methods produce good results (relative to e.g. the fine-tuning baselines) for larger numbers of shots. Can you point me to any results that support this?
> >
> > - RE: ‘suggestion’, that sounds like an interesting direction! I recommend taking a look at [1] as well which proposes to use gradients as features (though not in a meta-learning context).
> >
> > Additional comment
> > - Pushing more on the datasets side, why not actually use existing benchmarks for few-shot learning across datasets instead of creating a new one, in possible future iterations of this work? That would allow for direct comparisons with several models that are designed for the problem of heterogeneous data in few-shot learning and would also facilitate future work to compare against the approach proposed in this paper. Some options are the benchmark[1], the ‘multi-dataset’ assembled in [2], or Meta-Dataset.
> >
> > References
> > - [1] Gradients as Features for Deep Representation Learning. Mu et al. ICLR 2020.
> > - [2] A Broader Study of Cross-Domain Few-Shot Learning. Guo et al.
> > - [3] Learning to Balance: Bayesian Meta-learning for Imbalanced and Out-of-Distribution Tasks. Lee et al. ICLR 2020.

---

> > > ### Author Response · Authors · 2021-11-23
> > > **Reply to Reviewer 9y7H (1/2)**
> > >
> > > Dear Reviewer,
> > >
> > > Thank you for asking the questions. We have provided our responses as follows.
> > >
> > > * Please note that in Tian et al. (2020), MAML's accuracy of 48.7% and 63.1% is based on 4-layer convnet (with 32 filters at each layer), while their results for Finetune (termed "Ours-simple" in their paper) and Finetune-distill (termed "Ours-distill" in their paper) are based on ResNet-12 (see Table 1 in their paper, which contains a "backbone" column). Following the original implementation of MAML, our results on mini-ImageNet (Appendix D.1) are also based on 4-layer convnet. Hence, the large margin between the performance of Finetune in Tian et al. (2020) and MAML (as well as our CTML) could be due to the different choice of backbone. Based on our experiments with a deeper backbone, replacing 4-layer convnet with ResNet-12 indeed results in some performance gains for MAML and all the MAML-based baseline methods, though the performance gains for Finetune methods are relatively more significant when using a deeper backbone.
> > >
> > >   In fact, a better reference will be the results reported in Chen et al. (2019), where the authors implement Finetune (termed "Baseline" in their paper) and several well-known meta-learning algorithms (including MAML) for different backbones (i.e., conv-4, conv-6, ResNet-10, ResNet-18, ResNet-34), and compare the performance of different algorithms under the same backbone. Looking at their results on mini-ImageNet 1-shot scenario (see Table 2 and Figure 3 in their paper), MAML generally outperforms Finetune across different backbones, with a larger margin on shallower backbone (i.e., conv-4), and a smaller margin on deeper backbone (i.e., ResNet). Their results also demonstrate that increasing the depth of the backbone generally improves the performance of all the evaluated methods. Our experimental results are very much in line with their conclusions.
> > >
> > > * We did not come across many works that explicitly study the effectiveness of MAML-like methods on tasks with very large support set size. But the Meta-Dataset benchmark (Triantafillou et al. 2020) actually adopts a very interesting task/episode sampling strategy where the support set size ranges from some small number to 500 depending on the sub-dataset from which the task is sampled. Specifically, they determine the support set size of a sampled task following the formula $|\mathcal{S}|=\text{min}\\{500,\sum_{c\in\mathcal{C}}\lceil\beta\text{min}\\{100,|Im(c)|-q\\}\rceil\\}$, where $|Im(c)|$ denotes the number of images belonging to class $c$ in the chosen sub-dataset, $q$ is the query set size (determined by another formula but is capped at 10), and $\beta$ is a fraction sampled from $(0,1]$. This means that for a sub-dataset whose $|Im(c)|$ for each class is larger than 100, it will definitely contribute 100 images per class for sampling, and the resultant number of shots per class will be from 1 to 100 (i.e. $\lceil\beta \times100\rceil$). Also, they specify that the number of classes per task is in the range $[5,\text{total number of classes of that sub-dataset}]$, which means that the support set size can easily reach the cap of 500 for sub-datasets with a large number of classes and a large number of images per class. In fact, many of the sub-datasets they've used in their Meta-Dataset are large and very often produce near-500 support set size, including ImageNet, Fungi, Quick Draw, CUB Birds, and Textures.
> > >
> > >   In their experiments, they include both Finetune and the first-order MAML (i.e., fo-MAML) for comparison. They further propose a method termed fo-Proto-MAML which achieves the best results in their benchmark. This strong baseline is very similar to MAML except that it uses a task-specific classification layer initialized from the Prototypical Network-equivalent weights and biases (you can refer to their paper for detailed description). Regarding the number of fine-tuning/adaptation steps, based on their best-tuned hyperparameters settings, the number of fine-tuning steps for Finetune is 200, and the number of adaptation steps for fo-MAML and fo-Proto-MAML is 10 and 6 respectively. Under these best settings, they report that the performance of the 3 models follows: fo-Proto-MAML > Finetune > fo-MAML (see Table 1 in their paper). Given that the tasks sampled most often have large support set size, their results should be good evidence of how MAML-based methods can also outperform Finetune for large support set with just a few adaptation steps. Besides, their experiments also demonstrate that Finetune generally requires a large number of steps when adapting to new tasks (which we've also observed in our experiments of Finetune), making it a less preferred option when the speed of online task adaptation is of concern.

---

> > > > ### Comment · Reviewer_9y7H · 2021-11-23
> > > > **Reply to authors (round 2)**
> > > >
> > > > Thank you for the quick reply and additional discussion.
> > > >
> > > > - In the results reported in Chen et al. (2019), indeed MAML performs better than Baseline (in Table 2), but Baseline++ outperforms both of these, even with a conv-4 architecture (and, as you said, these fine-tuning methods actually benefit more than meta-learning methods by the increase in backbone size). This contradicts the results you reported in the rebuttal, where Baseline++ ('Fine-tune cosine distance') performs significantly worse than MAML, right? Of course, your results are on a different benchmark involving several datasets, whereas those in Chen et al. (2019)'s Table 2 are in the single-dataset setting, so they're not directly comparable.
> > > >
> > > > - Thank you for bringing up the results from the Meta-Dataset paper, this is indeed an interesting data point in this discussion. While their proto-MAML model did outperform other baselines in their investigation, they achieved this by initializing the feature extractor at the start of proto-MAML's meta-learning phase to a pre-trained ('Baseline') feature extractor. So this model is actually a hybrid between the Baseline(++) and a meta-learning model. Another relevant finding from that work is that similar results to Proto-MAML can be obtained via the 'inference-only proto-MAML' model, which actually is just the Baseline model, acting as a Proto-MAML *only* at test time (see Figure 3c and 3d of their paper and the 'Effect of meta-training' section). This means that the success of Proto-MAML there isn't necessarily due to being meta-learned, but is largely due to its useful way of tackling tasks at test time. Taken together, these observations suggest that MAML-like models don't necessarily perform significantly better than fine-tuning baselines in that context either.
> > > >
> > > > - Overall, the literature for both single-dataset and heterogeneous benchmarks shows that Baseline performs strongly. Given this, my main point here is to ensure that the baselines reported in the rebuttal are implemented correctly, since I was very surprised that even vanilla MAML in your results outperforms these strong baselines on the meta-dataset compiled in this work.
> > > >
> > > > - I agree with your point that fine-tuning typically requires significantly more adaptation steps in each test task compared to meta-learning methods, and this is indeed a nice advantage of meta-learning approaches.

---

> > > > > ### Author Response · Authors · 2021-11-25
> > > > > **Response to Reviewer 9y7H**
> > > > >
> > > > > Dear Reviewer,
> > > > >
> > > > > Thank you for the further discussion.
> > > > >
> > > > > For the proto-MAML (as well as other baselines) in the Meta-Dataset paper, they treated the choice of starting from scratch or starting from a ImageNet pre-trained initialization as a hyperparameter, since using a pre-trained model may or may not increase the performance. In Figure 3(a), they study the effect of using an ImageNet pre-trained initialization. If you scrutinize the results of proto-MAML carefully (compare the two bars for "pre-trained" and "from scratch" for every sub-dataset), you can see that actually using a pre-trained model does not add advantages for proto-MAML in general, which means that the superiority of proto-MAML comes from its own algorithmic design rather than relying on the pre-trained initialization.
> > > > >
> > > > > Nevertheless, the point we are trying to establish here is just that even for a larger support set size, MAML-like methods can still produce good results (at least comparable to the Finetune methods, if not significantly outperform) with a smaller number of adaptation steps, as a response to your previous concern about our argument that "for larger support set sizes, the number of inner loop steps can be still kept small and MAML-like methods are ‘still being able to find a good initialization."
> > > > >
> > > > > Regarding our experiments on the additional Finetune baselines, we adopted the implementations directly from the source codes released by the authors. It is possible that fluctuations occur when we are using a different dataset. Another reason for the lower than expected performance for the Finetune methods in our experiments is that we constrain the number of finetuning steps to 50 and tune the learning rate and optimizer, so as to allow a fairer comparison with the MAML-based baselines within a similar time budget at task adaptation phase. Allowing further steps until convergence will bring about 2~3% more performance gains for the Finetune methods.

---

> > > > > > ### Comment · Reviewer_9y7H · 2021-11-25
> > > > > > **response to authors**
> > > > > >
> > > > > > Thanks for the response. This change in hyperparameters might be largely responsible for the unexpected results. In my experience these hyperparameters (number of fine-tuning steps, learning rate and optimizer) can make a big difference, so it's important to report results for these baselines with the original design choices, to compare to a strong version of these.
> > > > > >
> > > > > > Some additional comments on computational efficiency: it should be noted that, while MAML methods take only a small number of steps in each test episode, they fine-tune the entire feature extractor for these steps, whereas these fine-tuning baselines (at least in Chen et al, and Tian et al) only train a new linear layer on top of a frozen feature extractor (so, in fact the name 'fine-tuning' is misleading, I wish they were called 'linear probing' instead, or something else). This means that it's perhaps reasonable to allow a larger step budget, since only a few readout parameters are trained. Furthermore, these baselines are significantly more memory and computation efficient at *meta-training* time, especially compared to second-order (vanilla) MAML and its variants which require episodic training and second-order derivatives. So there is a trade-off between efficiency at training time versus deployment time, and for different applications one choice might be more appropriate than the other.
> > > > > >
> > > > > > RE: Meta-Dataset results, you're right that for Proto-MAML in particular the results are somewhat mixed with respect to the importance of the initialization, but what I mentioned previously is still true, that the 'Proto-MAML inference baseline' (i.e. applying Proto-MAML's algorithm only at test time, on a pre-trained feature extractor) performs similarly most of the time compared to Proto-MAML (this is especially true when training happens on 'all datasets' instead of only ImageNet). I think more work is needed to understand the usefulness of meta-learning a feature extractor for few-shot learning. In the meantime, I don't think we have very strong indications that it is (very) beneficial for few-shot classification at least.
> > > > > >
> > > > > > More generally, the proposed model seems like a nice meta-learning variant that outperforms similarly-motivated work in the case of heterogeneous data, and I agree with the authors that there are cases where computational efficiency at test time is important and perhaps this is a scenario where meta-learning approaches are preferable. However, the simpler methodology of pre-training, followed by linear probing or fine-tuning, forms the basis for the methods that are now state-of-the-art, both in single-dataset as well as heterogeneous benchmarks (happy to provide more references for the latter). So I think it's important to show a comparison to this simple methodology (with appropriate hyperparameters for these baselines), and discuss the pros and cons of these methods.

---

> > > > > > > ### Author Response · Authors · 2021-11-26
> > > > > > > **Response to Reviewer 9y7H**
> > > > > > >
> > > > > > > Dear Reviewer,
> > > > > > >
> > > > > > > Thank you for the very prompt reply!
> > > > > > >
> > > > > > > We certainly agree with your point that "there is a trade-off between efficiency at training time versus deployment time, and for different applications one choice might be more appropriate than the other".  We therefore conducted further experiments to allow for an unconstrained number of fine-tuning steps until convergence (and it is around 500 steps in our best hyperparameters setting).
> > > > > > >
> > > > > > > Below are the best results we've obtained for the Finetune methods. For deeper backbone ResNet-12, the two Finetune variants indeed outperform MAML, while for Conv-4, there are still some gaps between the Finetune methods and MAML. Compared to the results in Chen et al. (2019) (Figure 3 mini-ImageNet 1shot), though Baseline++ outperforms MAML for Conv-4, it underperforms MAML for a slightly deeper backbone Conv-6. Hence, we believe that the results we obtained here fall within reasonable fluctuations.
> > > > > > >
> > > > > > > | Methods                                     | Conv-4 | ResNet-12 |
> > > > > > > | ------------------------------------------- | ------ | --------- |
> > > > > > > | Finetune                                    | 49.01  | 56.19     |
> > > > > > > | Finetune-cosine distance (Chen et al. 2019) | 50.64  | 57.59     |
> > > > > > > | Finetune-distill (Tian et al. 2020)         | 51.03  | 58.82     |
> > > > > > > | MAML                                        | 52.83  | 56.84     |
> > > > > > > | MMAML                                       | 54.28  | 59.12     |
> > > > > > > | L2F                                         | 56.61  | 60.69     |
> > > > > > > | HSML                                        | 56.32  | 61.57     |
> > > > > > > | ARML                                        | 57.21  | 61.06     |
> > > > > > > | CTML (ours)                                 | 59.03  | 63.18     |
> > > > > > >
> > > > > > > We will revise the corresponding section in our paper to reflect the best results shown here, and add some discussion about the pros and cons of these simple Finetune methods, as suggested in your comments.
> > > > > > >
> > > > > > > About the Meta-Dataset paper, the findings of the "inference-only" meta-learning methods are indeed intriguing. But the vanilla MAML is not studied explicitly in this experiment as the initialization of the final classification layer has to be meta-learned (pre-trained model has classification layer that outputs a different dimensionality). For other MAML variants like our method and the baselines we've used in our experiments, we all have parameterized modules (e.g., modulation network, path learner, clustering structure like k-means centroids in our case, hierarchal tree in HSML, and meta-knowledge graph in ARML) that have to be meta-learned to abstract the transferrable knowledge from tasks. And since these designs indeed boost performance in the heterogeneous setting, we believe that the meta-training phase is still essential.
> > > > > > >
> > > > > > > Everything aside, we would like to take the time to thank you for your valuable advice and your willingness to engage in this very interesting and enlightening discussion with us! It is undoubtedly an enjoyable experience and will definitely help us in our future research in meta-learning and few-shot learning!

---

> > > > > > > > ### Comment · Reviewer_9y7H · 2021-11-26
> > > > > > > > **response to authors**
> > > > > > > >
> > > > > > > > Thank you for running these results so quickly. 500 steps sounds more appropriate for these baselines and these results are more in line with what I was expecting. It would be great to also run these for 5-shot 5-way too (I think these results are 1-shot?).
> > > > > > > >
> > > > > > > > It is still interesting that for conv-4 even vanilla MAML is better than Baseline++ (contrary to Chen et al). One hypothesis for this is that, because of the heterogeneity of this benchmark, the ability to actually adapt the feature extractor (which MAML does but Baseline++ does not) is more important than it was for the datasets explored in (Chen et al). This hypothesis can be examined in two ways:
> > > > > > > > - 1) Allow the Fine-tuning baselines to also adapt the feature extractors in each test task (instead of training only a readout layer). In the Meta-Dataset paper this was found to work better so perhaps the same is true for this heterogeneous benchmark too, which would constitute evidence in support of this hypothesis
> > > > > > > > - 2) Take the feature extractor trained by vanilla MAML and at test time treat it like a Baseline++ where only the readout layer is trained for each new test task. If the results for this are substantially worse than the MAML results, this also would be evidence in support of this hypothesis.
> > > > > > > >
> > > > > > > > But this additional exploration is now perhaps derailing too much from the paper's contributions and CTML.
> > > > > > > >
> > > > > > > > Yes, it's true that some meta-learners meta-learn other components too, aside from the feature extractor (like in the examples you provided), and in this case it's not possible to compute the 'inference-only' counterpart in the same way. But in these cases, it's still unclear whether meta-learning is always beneficial for the feature extractor itself. Instead, it may be possible to keep a pre-trained feature extractor frozen throughout the meta-learning phase and only meta-learn the additional components. This is what I had in mind in the suggestion in my initial review, where I referred to the setup in CNAPs (Requeima et al, NeurIPS 2019), so I think we've come full circle :)

---

> > > > > > > > > ### Author Response · Authors · 2021-11-30
> > > > > > > > > **Response to Reviewer 9y7H**
> > > > > > > > >
> > > > > > > > > Dear Reviewer,
> > > > > > > > >
> > > > > > > > > In the past few days, we conducted further experiments to tune the Finetune methods more carefully (i.e., tune the pre-training process, save the pre-training checkpoints more frequently and evaluate/fine-tune the models). It turns out that the Finetune methods are able to outperform vanilla MAML for Conv-4 as well! The two tables below present the 1-shot and 5-shot results respectively.
> > > > > > > > >
> > > > > > > > > | Methods (1-shot)                            | Conv-4 | ResNet-12 |
> > > > > > > > > | ------------------------------------------- | ------ | --------- |
> > > > > > > > > | Finetune                                    | 51.22  | 58.02     |
> > > > > > > > > | Finetune-cosine distance (Chen et al. 2019) | 52.98  | 59.13     |
> > > > > > > > > | Finetune-distill (Tian et al. 2020)         | 53.64  | 59.70     |
> > > > > > > > > | MAML                                        | 52.83  | 56.84     |
> > > > > > > > > | MMAML                                       | 54.28  | 59.12     |
> > > > > > > > > | L2F                                         | 56.61  | 60.69     |
> > > > > > > > > | HSML                                        | 56.32  | 61.57     |
> > > > > > > > > | ARML                                        | 57.21  | 61.06     |
> > > > > > > > > | CTML (ours)                                 | 59.03  | 63.18     |
> > > > > > > > >
> > > > > > > > > | Methods (5-shot)                            | Conv-4 | ResNet-12 |
> > > > > > > > > | ------------------------------------------- | ------ | --------- |
> > > > > > > > > | Finetune                                    | 65.08  | 71.11     |
> > > > > > > > > | Finetune-cosine distance (Chen et al. 2019) | 67.42  | 72.23     |
> > > > > > > > > | Finetune-distill (Tian et al. 2020)         | 68.17  | 72.20     |
> > > > > > > > > | MAML                                        | 66.54  | 69.84     |
> > > > > > > > > | MMAML                                       | 67.82  | 71.92     |
> > > > > > > > > | L2F                                         | 68.45  | 72.07     |
> > > > > > > > > | HSML                                        | 68.73  | 73.35     |
> > > > > > > > > | ARML                                        | 69.29  | 73.56     |
> > > > > > > > > | CTML (ours)                                 | 72.18  | 75.03     |
> > > > > > > > >
> > > > > > > > > Indeed, with careful hyperparameters tuning, the Finetune methods are very strong baselines! They even become comparable to some of the task-conditioned methods like MMAML and L2F. (we want to apologize for updating the results of the Finetune methods several times, as we do not have much experience with the Finetune methods, and the hyperparameters tuning is kinda tricky. So far these are the best results we can get.)
> > > > > > > > >
> > > > > > > > > Regarding your suggestions to also finetune the feature extractor, we think that it is a great way to see if this will help in the heterogeneous setting, especially for the shallower backbones like Conv-4, which have lower capacity to capture the diversity during pre-training on the heterogeneous datasets. However, based on our few trials of experiments to also finetune the feature extractor, we did not observe evident performance gains for both Conv-4 and ResNet-12. Perhaps with further hyperparameters tuning we will be able to see some improvements (the pre-trained feature extractor will need a much lower learning rate to finetune as compared to the readout layer). We will continue to try :)
> > > > > > > > >
> > > > > > > > > The two-stage process of first pre-training a feature extractor and then keeping it frozen while learning the meta components sounds like a very promising direction! We will definitely look further into it in our future investigations regarding task-conditioned meta-learning with optimization-based representation.

---

> > > > > > > > > > ### Comment · Reviewer_9y7H · 2021-11-30
> > > > > > > > > > **response to authors**
> > > > > > > > > >
> > > > > > > > > > Thank you for the additional results, these are indeed more convincing for the fine-tuning baselines.
> > > > > > > > > >
> > > > > > > > > > RE: Meta-Dataset, which architecture was used for these experiments? If it's resnet-12, perhaps simply switching to resnet-18 will improve further. It would be great to also report the performance on all datasets (instead of just the OOD ones) in the revised version of the paper. It's possible that models like SUR and URT outperform CMTL on the in-distribution datasets, but CTML has the advantage of parameter efficiency compared to these approaches which train a separate feature extractor per training dataset. A more fair comparison is perhaps to their 'parameteric family' variants SUR-pf, and URT-pf, as well as to FLUTE which has a comparable parameter count too.
> > > > > > > > > >
> > > > > > > > > > Note on traffic signs dataset: there were two versions of this accidentally due to a bug (https://github.com/google-research/meta-dataset/issues/54). The intended (and harder) version shuffles the data, whereas the other one uses an unshuffled dataset. So it is important for consistency of comparisons between models to report results on the same version in the table. e.g. the results you report for SUR above are on the easier version whereas for URT, FLUTE and URL are on the harder version. The results on the intended shuffled version of the datasets can be found on the public leaderboard, in the README section of the repo (https://github.com/google-research/meta-dataset) and sometimes are also reported in the appendix of papers. It would be great to clarify which version was used to obtain the results reported here for CTML too.
> > > > > > > > > >
> > > > > > > > > > Generally, recent methods like SUR, FLUTE, URL do not use any meta-learning. URT meta-learns only an attention mechanism to select features on top of pre-trained (frozen) backbones. So reporting strong performance on this benchmark with CTML would be an interesting data point in favour of meta-learning approaches. You might also be interested in the newer benchmark (Dumoulin et al) that combines Meta-Dataset with the VTAB benchmark for transfer learning, and in the discussions there about transfer vs meta-learning methods, as well as the recent discussion and results on scaling up meta-learners in (Bronskill et al).
> > > > > > > > > >
> > > > > > > > > > I have updated my score to a 6, to reflect that throughout these discussions you have addressed several of my concerns (relationship to baselines, ablations, deeper backbones). I think the preliminary results reported here and directions discussed are promising for further strengthening the paper in the future.
> > > > > > > > > >
> > > > > > > > > > References:
> > > > > > > > > > - Dumoulin et al. Comparing Transfer and Meta Learning Approaches on a Unified Few-Shot Classification Benchmark. NeurIPS 2021.
> > > > > > > > > > - Bronskill et al. Memory Efficient Meta-Learning with Large Images. NeurIPS 2021.

---

> > > > > > > > > ### Author Response · Authors · 2021-11-30
> > > > > > > > > **Response to Reviewer 9y7H**
> > > > > > > > >
> > > > > > > > > **Results on Public Benchmark**
> > > > > > > > >
> > > > > > > > > As mentioned earlier, we have conducted additional experiments on the public Meta-Dataset benchmark [1]. The table below presents our results on the out-of-domain evaluation datasets against some of the most recent state-of-the-art methods. Though not significantly outperforming the state-of-the-art, our method is able to achieve very competitive performance.
> > > > > > > > >
> > > > > > > > > |              | Proto-MAML [1] | BOHB-E [2] | CNAPS [3] | TaskNorm [4] | Simple CNAPS [5] | SUR [6]     | URT [7]   | URL [8]  | FLUTE [9] | Ours     |
> > > > > > > > > | ------------ | -------------- | ---------- | --------- | ------------ | ---------------- | ----------- | --------- | -------- | --------- | -------- |
> > > > > > > > > | Traffic Sign | 52.42 (9)      | 57.61 (8)  | 60.2 (6)  | 67 (4)       | 73.5 (1)         | 70.4   (2)  | 51.1 (10) | 63.3 (5) | 58.4 (7)  | 69.7 (3) |
> > > > > > > > > | MSCOCO       | 41.74 (10)     | 51.86 (5)  | 42.6 (9)  | 43.4 (8)     | 46.2 (7)         | 52.4    (3) | 52.2  (4) | 54 (1)   | 50 (6)    | 52.6 (2) |
> > > > > > > > > | MNIST        | -              | -          | 92.7 (7)  | 92.3 (8)     | 93.9 (6)         | 94.3 (5)    | 94.8 (3)  | 94.7 (4) | 95.6 (1)  | 95.4 (2) |
> > > > > > > > > | CIFAR-10     | -              | -          | 61.5 (7)  | 69.3 (4)     | 74.3 (2)         | 66.8 (6)    | 67.3 (5)  | 74.2 (3) | 78.6 (1)  | 67.3 (5) |
> > > > > > > > > | CIFAR-100    | -              | -          | 50.1 (8)  | 54.6 (7)     | 60.5 (4)         | 56.6 (6)    | 56.9  (5) | 63.5 (2) | 67.1 (1)  | 61(3)    |
> > > > > > > > > | Avg. Rank    | 9.5            | 6.5        | 7.4       | 6.2          | 4                | 4.4         | 5.4       | 3        | 3.2       | 3        |
> > > > > > > > >
> > > > > > > > > [1] Triantafillou, Eleni, et al. "Meta-Dataset: A Dataset of Datasets for Learning to Learn from Few Examples." *ICLR*. 2019.
> > > > > > > > >
> > > > > > > > > [2] Saikia, Tonmoy, Thomas Brox, and Cordelia Schmid. "Optimized generic feature learning for few-shot classification across domains." *arXiv.* 2020.
> > > > > > > > >
> > > > > > > > > [3] Requeima, James, et al. "Fast and flexible multi-task classification using conditional neural adaptive processes." *NeurIPS*. 2019.
> > > > > > > > >
> > > > > > > > > [4] Bronskill, John, et al. "Tasknorm: Rethinking batch normalization for meta-learning." *ICLR*. 2020.
> > > > > > > > >
> > > > > > > > > [5] Bateni, Peyman, et al. "Improved few-shot visual classification." *CVPR*. 2020.
> > > > > > > > >
> > > > > > > > > [6] Dvornik, Nikita, Cordelia Schmid, and Julien Mairal. "Selecting relevant features from a multi-domain representation for few-shot classification." *ECCV*. 2020.
> > > > > > > > >
> > > > > > > > > [7] Liu, Lu, et al. "A Universal Representation Transformer Layer for Few-Shot Image Classification." *ICLR*. 2020.
> > > > > > > > >
> > > > > > > > > [8] Li, Wei-Hong, Xialei Liu, and Hakan Bilen. "Universal Representation Learning from Multiple Domains for Few-shot Classification." *ICCV*. 2021.
> > > > > > > > >
> > > > > > > > > [9] Triantafillou, Eleni, et al. "Learning a Universal Template for Few-shot Dataset Generalization." *ICML*. 2021.

---

> > > ### Author Response · Authors · 2021-11-23
> > > **Reply to Reviewer 9y7H (2/2)**
> > >
> > > * Thank you for pointing out this work! It is indeed very interesting and relevant to our current focus.
> > >
> > > * We've created a new collection of datasets because for all the task-conditioned baselines we've chosen, they are using different combinations of datasets to simulate task heterogeneity. We basically combined their choices of sub-datasets and constructed a new one for our experiments. In fact, we are not aware of the existence of the Meta-Dataset benchmark until you pointed it out (since none of the relevant works we've surveyed conducted experiments on this benchmark, partly because it is a very recent dataset). Nevertheless, we would like to thank you for pointing out this good evaluation benchmark. We are currently working on experiments with this Meta-Dataset benchmark, and since it is a very large collection of datasets, it requires some time to pre-process the data and conduct meta-training. We will post the results here as long as we've completed the experiments.

---

> ### Author Response · Authors · 2021-11-18
> **Response to Reviewer 9y7H (2/3)**
>
> **3. Scalability considerations**
>
> - Scalability with deeper backbone
>
>   Like all the other task-conditioned MAML-based methods (the baselines we've compared in the paper), deeper backbone inevitably leads to increased space and time complexity of the proposed meta-model. This is because larger parameter size of the backbone will require a larger 'modulation network' to generate the element-wise masks on the global initialization, and deeper backbone will result in slower task adaptation as now we have more layers to back-propagate the gradients. However, luckily, **all the *new* innovations and modules we've introduced in our paper have parameter sizes that are *independent* of the size of the backbone**.
>
>   Looking at Figure 1 (overview of our proposed framework), our main contributions are part (a) which models the learning path, and part (d) which creates a mapping between the feature and path cluster assignments to provide a shortcut at inference time.
>
>   For part (a), the same GRU path learner (described in equation 3) is used to process the path tuple $\textbf{P}^d_{\mathcal{T}_i}\in\mathbb{R}^{(\tau+1)\times 4}$ at all dimensions of the backbone $d\in \\{1, 2, ..., D\\}$, where $D$ is the total parameter size of the backbone. This means that **the proposed GRU path learner is independent of the backbone size** (note that the GRU trainable weights $\textbf{W}_r,\textbf{W}_z,\textbf{W}_h$ are not subscripted by $d$). In fact, we have mentioned this in section 4.1.2, the last two sentences of the second paragraph.
>
>   For part (d), the 'shortcut tunnel' adopts a 2-layer fully connected network design to map from feature cluster assignments to path cluster assignments for all the tasks. Hence, **this 'shortcut' network is also not related to the backbone size**, and is only affected by the number of clusters for feature and path embeddings.
>
>   For part (b) and part (c), the concepts of feature representation and task-aware modulation are not new and are also adopted in the task-conditioned baselines. In fact, only the task-aware modulation network will have its size increased with the backbone, the size of the feature extractor in part (b) does not depend on the backbone (note that the feature extractor discussed here is a separate network to be learned, it is not the same as the feature extractor of backbone).
>
> - Scalability with more adaptation steps
>
>   Following the baselines, we set the number of adaptation steps as 5 for both few-shot classification and cold-start recommendation (we have stated this in Appendix B). Larger number of adaptation steps inevitably leads to slower task adaptation. However, this problem exists not just for our method but for all the MAML-based meta-learning algorithms. In fact, the main problem MAML tackles is faster and better adaptation of new tasks. That means we constrain the number of adaptation steps to a small number and search for a good initialization which allows for convergence within just these few steps. For larger support set size, we can still do the same by constraining the number of adaptation steps and still being able to find a good initialization.
>
>   As compared to the baselines, the major limitation of our method is probably the additional computational cost incurred by the rehearsed learning and the path modeling procedure. However, we are well-aware of this problem and have therefore proposed the 'shortcut tunnel' to bypass this process at online testing (i.e., the process depicted by part (a) will not be gone through if part (d) is applied). Hence, at meta-testing phase, our method with shortcut approximation has inference time comparable to the baselines (and sometimes faster), as shown in Table 1(a).
>
>   Regarding the concern over the increased computational cost at meta-training time (where the rehearsed learning cannot be bypassed), a way to alleviate this is to reduce the number of rehearsed adaptation steps. As observed from the rehearsed learning paths plot (Figure 3(b)) and the z and r gates of GRU path learner at various steps (Figure 3(c)), we can see that most often the first 1 to 3 steps are more important (i.e., with darker z gate) and contributing more information to the learning path characterization. After that the learning will converge (as we can see from the plot that the last few steps often vibrate about the same location), and the steps are not adding more information about the path (i.e., with lighter z gate). Based on this observation, we can reduce the number of rehearsed adaptation steps by a small amount without sacrificing a great deal of performance. In fact, we have conducted experiments to test the effect, and the results show that reducing the number of rehearsed steps by 1~2 will not cause significant drop in performance. Nevertheless, since the number of adaption steps is constrained to a small number for MAML-based methods, the cost at meta-training time should not be the main concern.

---

> ### Author Response · Authors · 2021-11-18
> **Response to Reviewer 9y7H (3/3)**
>
> **4. Additional ablation**
>
> To ablate the effect of clustering for both the path and feature embeddings, we introduce two variants: CTML-feat-nocluster and CTML-path-nocluster. The former is CTML with feature representation only and no clustering, and the latter is CTML with path representation only and no clustering. We compare them with CTML-feat and CTML-path respectively. The results on Meta-Dataset (the one created by us) are shown in the table below:
>
> |                     | 5-way 1-shot | 5-way 5-shot |
> | ------------------- | ------------ | ------------ |
> | CTML-feat-nocluster | 55.21        | 67.49        |
> | CTML-feat           | 57.14        | 68.71        |
> | CTML-path-nocluster | 56.20        | 69.11        |
> | CTML-path           | 57.74        | 70.63        |
>
> We can see that without clustering, the performance slightly drops for both the path and feature embeddings. This shows that performing clustering indeed helps to generalize better by explicitly considering the global structure. To see the effect of changing the number of clusters, please refer to section 5.2 and Figure 4(a) in the paper, where we test the sensitivity to the number of clusters/user groups for MovieLens dataset.
>
> **5. Some typos**
>
> Thank you so much for reading our paper so carefully and pointing out the typos! We will revise the parts accordingly. And about the "probe" network, it is the model that is responsible for performing the task (the term is adopted from [Achille et al. (2019)](https://arxiv.org/pdf/1902.03545.pdf)). For instance, in the case of Achille et al. (2019), the meta-learning objective is to select the best pre-trained feature extractor for a new task, hence the "probe" network is the pre-trained feature extractor to be evaluated at hand. In our case, the "probe" network is the base-learner adapted to tasks.
>
> **6. Suggestion**
>
> Regarding your suggestion to apply our ideas to the work of Requeima et al. (2019), we realize that it aims to meta-learn an adaptation network to generate the task-specific parameters based on the input dataset, and then adapt a pre-trained global feature extractor to individual tasks based on the task-specific parameters. One direct application of our idea is that, we can conduct few steps of task finetuning from the global feature extractor, and then input the path embedding obtained into the adaptation network together with the other task-specific features. However, we realize that this may be against the spirit of the amortized methods, which is to avoid slow adaptation. A compromised way will be to input only the gradients of the global feature extractor w.r.t. the task loss, which also serves to provide information from the optimization perspective. Nevertheless, this is just a rough idea that we think of for now. In fact, we plan to further investigate the benefits of representing tasks from the optimization perspective for more meta-learning approaches, including the metric-based and model-based/amortized methods.

---

### Official Review · Reviewer_fGEV · 2021-11-03

**Correctness:** 3
**Technical Novelty And Significance:** 3
**Empirical Novelty And Significance:** 3
**Recommendation:** 6
**Confidence:** 2

**Main Review:**

The proposed idea that adopts a learning path to improve the parameter initialization is interesting. Code is provided and experiment results look persuasive.

Some of the critical technical points are not sufficiently justified, e.g.
(1)	why is using the trajectory of rehearsed path helpful to learn a good parameter initialization?
(2)	in section 4.2, the reason for modeling task representation is explained as encoding the predictive distribution. But I can't see the connection between them.
  I would like the author to use several sentences to explain the insights of these points.


**Summary Of The Paper:**

This paper proposes a clustered task-aware meta-learning algorithm. The proposed algorithm firstly collects the learning path and uses the path to train a sequence module. The sequence module output is combined with the task feature. The task feature is derived from the weighted sum of the soft cluster centers. The combination is then used as initialization of the model parameter for task adaptation. Experiments on image classification and cold-start recommendation demonstrate superior performance compared to baseline algorithms.

**Summary Of The Review:**

The proposed algorithm is interesting and well-supported by experiments. My concerns are that a more intuitive description should be added to make the motivation, rationality, and insight of algorithm design easy to follow.

---

> ### Author Response · Authors · 2021-11-18
> **Response to Reviewer fGEV**
>
> Dear Reviewer:
>
> Thank you so much for your valuable comments! Regarding your concerns, we have provided our responses as follows.
>
> **1. How does rehearsed learning path help to learn a good initialization?**
>
> Firstly, rehearsed learning path helps to characterize the task from the optimization perspective. It provides additional information for better task representation on top of input features. The final representation aggregating both path and feature is therefore more informative about the task characteristics, and using it to modulate the initialization will enhance the inductive bias specific to the task.
>
> Secondly, the rehearsed learning path provides information about how well the task will learn with the base-model from the global initialization. It can be considered as encoding the quality of the initialization with respect to the task, as tasks with longer and more zigzag paths imply that the initialization is not a good one for them. By encoding this perspective, the path representation should provide a stronger inductive bias to modulate the initialization in the favorable direction that facilitates task learning.
>
> **2. On the connection between learning path and conditional distribution**
>
> Note that the learning path of a task is directed by the labels (i.e., learning proceeds in direction that optimizes the loss). Hence, unlike the feature embeddings, which if plotted for all the tasks in the latent space will form the marginal distribution $p(\mathbf{x})$, the path embeddings in the latent space will constitute the conditional distribution $p(y|\mathbf{x})$.
>
> To illustrate our point, here we give a simple example on binary classification problem. Suppose each data instance $i$ consists of a 2-D feature $\mathbf{x}^{(i)}=[x_1, x_2]$ and a label $y^{(i)}\in \\{0,1\\} $. We apply the logistic model $\hat{y}=\sigma(z)=\frac{1}{1+e^{-z}}$,  where $z=\beta_0+\beta_1x_1+\beta_2x_2$, and minimize the log loss $\mathcal{l}=-y\log(\hat{y})+(1-y)\log(\hat{y})$. To further simplify the case, we assume each task consists of only one instance (that is, each instance is treated as a task). Consider 3 tasks $\\{\mathcal{T}_1, \mathcal{T}_2, \mathcal{T}_3\\}$ with exactly the same feature $[x_1=2, x_2=4]$ but different label $y$, we can compute the derivatives of $l$ w.r.t. $\beta_0$, $\beta_1$ and $\beta_2$ as follows:
>
> | $\mathcal{T}_i$ | $x_1$ | $x_2$ | $y$  | $\frac{\partial{l}}{\partial{\beta_0}}$ | $\frac{\partial{l}}{\partial{\beta_1}}$ | $\frac{\partial{l}}{\partial{\beta_2}}$ |
> | --------------- | ----- | ----- | ---- | --------------------------------------- | --------------------------------------- | --------------------------------------- |
> | 1               | 2     | 4     | 0    | $\frac{1}{e^z+1}$                       | $\frac{2}{e^z+1}$                       | $\frac{4}{e^z+1}$                       |
> | 2               | 2     | 4     | 0    | $\frac{1}{e^z+1}$                       | $\frac{2}{e^z+1}$                       | $\frac{4}{e^z+1}$                       |
> | 3               | 2     | 4     | 1    | $\frac{-1}{e^z+1}$                      | $\frac{-2}{e^z+1}$                      | $\frac{-4}{e^z+1}$                      |
>
> Since the learning path follows the gradient descent direction, we can see that $\mathcal{T}_1$ and $\mathcal{T}_2$ will have exactly the same learning paths, while $\mathcal{T}_3$ will have a learning path that points in opposite direction. Hence, the path embeddings generated for $\mathcal{T}_1$ and $\mathcal{T}_2$ will be positioned together in the latent space, while $\mathcal{T}_3$ will be positioned far apart. The overall 3-data point distribution in the latent space essentially describes the conditional distribution where $p(y=0|x_1=2, x_2=4)=\frac{2}{3}$ and $p(y=1|x_1=2, x_2=4)=\frac{1}{3}$. On the other hand, the feature embeddings which are not informed by the labels will position all the 3 tasks together, representing the marginal distribution $p(x_1=2, x_2=4)=1$.

---

### Author Response · Authors · 2021-11-23
**Revision Uploaded**

We thank all reviewers for their valuable comments. We have submitted a revised paper with the following changes:

Address to Reviewer 1:

1) In section 4.1.1, the last paragraph, added "Furthermore, leveraging path representations serves to ... " to elaborate on how can the rehearsed learning path help to learn a good initialization.
2) In section 4.2, the 1st paragraph, added "as the learning path is informed by the labels" in parenthesis to explain the connection between learning path and conditional distribution.

Address to Reviewer 2:

1. In appendix D.1, added experiments to compare with non-MAML baselines on a deeper backbone ResNet-12.
2. In section 2 (related work), the 1st paragraph, included (Requeima et al. 2019) and (Bateni et al. 2020) when discussing task-conditioned methods.
3. In section 5.1 (experiments on few-shot classification), renamed "Meta-Dataset" to "Mixture-Of-Datasets".
4. In section 3, the last paragraph, rephrased "Though task-specific methods ..., they overlook generalization among similar tasks" to "Though task-specific methods ..., they lack explicit modeling of the global clustering structure".
5. In section 5.1, the 2nd last paragraph (ablation study) and table 2(b), added experimental results of removing clustering for both path and feature.
6. In section 1, the 3rd paragraph, rephrased "The amount of information ... about a 'probe' network" to "The amount of information ... about a network that is responsible for performing the task".
7. In section 2, the 2nd paragraph, removed "not" from "... is often not neglected".
8. In section 2, the 3rd paragraph, added "e.g., at the initialization" in parenthesis to exemplify what the single point can be in the previous work.
9. In section 4, the 1st paragraph, added "on" after "elaborate".

Address to Reviewer 3:

1. In section 2 (related work), the 3rd paragraph, included the work of (Garcia, Jezabel R., et al. 2021) when discussing methods that consider gradients or learning trajectories for task representations.
2. In section 4.4, the 2nd paragraph, revised the sentence explaining the assumption from "tasks with similar feature assignments will also have similar path assignments" to "there exists a one-to-one mapping between the feature cluster assignments and the path cluster assignments, which can be linear or non-linear".
3. In section 4.1.1, the 2nd paragraph, removed the discussion about natural gradients.
4. In section 4.2, the 1st paragraph, added "as the learning path is informed by the labels" in parenthesis to explain the connection between learning path and conditional distribution.
5. Throughout the paper, rephrased "lower-order / higher-order geometric quantities" to "lower-order / higher-order learning path behaviors"
6. In section 5.1 (experiments on few-shot classification), renamed "Meta-Dataset" to "Mixture-Of-Datasets".

Address to Reviewer 4:

1. In section 4.4, the 2nd paragraph, revised the sentence explaining the assumption from "tasks with similar feature assignments will also have similar path assignments" to "there exists a one-to-one mapping between the feature cluster assignments and the path cluster assignments, which can be linear or non-linear".
2. In appendix D.1, added experiments to compare with baselines on a deeper backbone ResNet-12.

---

### Decision · Program_Chairs · 2022-01-20

**Decision:**

Reject

**Comment:**

This paper proposed a cluster-based task-aware meta-learning (CTML) approach with task representation learned from its own learning path. Based on the prior work of feature-based task characterization, it integrates the rehearsed task gradient descent trajectory into task representation, and further improves computational efficiency by learning a different network to estimate the rehearsed task-trajectory characterization from the feature representation. Experiments were conducted on few-shot image classification (meta dataset and miniimagenet) and cold-start recommendation tasks.

Reviewers had raised various concerns about the work including technical novelty, the shortcut-tunnel assumption, empirical comparison, scalability issue, more ablations for in-depth analysis, etc. The reviewers and AC appreciate authors for putting good efforts in the rebuttal by replying the review questions carefully and making changes to improve their experiments and paper.

Overall, this paper is a borderline case, where reviewers agree some clear merits (well-written, easy to follow, good execution of an interesting idea with code provided, etc). Despite the improvements during rebuttal, some major concerns on the weaknesses still remain (e.g., technical novelty, more convincing justification on the assumption, significance of empirical gains). Therefore, I cannot recommend it for acceptance at its current form, but I hope to see it accepted in the near future after these issues are fully addressed.